# Promiscuity of response regulators for thioredoxin steers bacterial virulence

Ju-Sim Kim [1,5], Alexandra Born[2,5], James Karl A. Till [1], Lin Liu [1], Sashi Kant [1], Morkos A. Henen[2,3], Beat Vögeli [2] & Andrés Vázquez-Torres[1,4] ✉

The exquisite specificity between a sensor kinase and its cognate response regulator ensures faithful partner selectivity within two-component pairs concurrently firing in a single bacterium, minimizing crosstalk with other members of this conserved family of paralogous proteins. We show that conserved hydrophobic and charged residues on the surface of thioredoxin serve as a docking station for structurally diverse response regulators. Using the OmpR protein, we identify residues in the flexible linker and the C-terminal β-hairpin that enable associations of this archetypical response regulator with thioredoxin, but are dispensable for interactions of this transcription factor to its cognate sensor kinase EnvZ, DNA or RNA polymerase. Here we show that the promiscuous interactions of response regulators with thioredoxin foster the flow of information through otherwise highly dedicated two-component signaling systems, thereby enabling both the transcription of *Salmonella* pathogenicity island-2 genes as well as growth of this intracellular bacterium in macrophages and mice.

Two-component systems dominate prokaryotic and archaeal signaling, and are widely utilized by eukaryotic organisms as diverse in the phylogenetic tree as slime molds, yeasts, and plants. In a typical two-component system[1], a histidine residue in the sensor kinase is autophosphorylated in response to environmental stimuli, which in pathogenic organisms such as *Salmonella* include acidity, osmolarity, short-chain fatty acids or magnesium[2]. The phosphoryl group is transferred from the histidine kinase to a highly conserved aspartic acid residue in the N-terminal receiver domain of the cognate response regulator[3] (Fig. S1a). The selectivity afforded by a few amino acids in the interfacial α-helices of a sensor kinase and its response regulator pair guarantees the transmission of specific signaling outputs, minimizing crosstalk with homologous two-component systems simultaneously transmitting signals within a single cell[4–7]. Phosphorylation of the aspartic acid triggers homo-dimerization of response regulators, exposing DNA-binding α-helixes or β-hairpins in the C-terminal effector domain. Response

regulators recruit RNA polymerase to promoters or counter-silence nucleoid proteins as has been described for OmpR or PhoP, respectively[8–11]. RNA polymerase transcribes genes encoding downstream adaptive outputs. In *Salmonella*, for instance, the response regulators OmpR, PhoP, and SsrB transcriptionally orchestrate virulence programs that increase the fitness of this enteric pathogen in gut and systemic sites[12–17].

SsrB is the master activator of the horizontally-acquired *Salmonella* pathogenicity island-2 gene cluster that encodes a secretion apparatus and translocon, as well as chaperones and effectors of a dedicated type III secretion system, expressed intracellularly in epithelial cells and macrophages[18–20]. SsrB also plays an important role in the regulation of ancestral genes[19]. In addition to canonical control via phosphorylation[21], SsrB is under allosteric control by the ancestral protein thioredoxin[22]. The atomic mechanisms that enable the association of thioredoxin to SsrB have not yet been revealed; however, binding of thioredoxin to SsrB can occur independently of canonical

[1]University of Colorado School of Medicine, Department of Immunology & Microbiology, Aurora, Colorado, USA. [2]University of Colorado School of Medicine, Department of Biochemistry & Molecular Genetics, Aurora, Colorado, USA. [3]Faculty of Pharmacy, Mansoura University, Mansoura 35516, Egypt. [4]Veterans Affairs Eastern Colorado Health Care System, Denver, Colorado, USA. [5]These authors contributed equally: Ju-Sim Kim, Alexandra Born. ✉e-mail: Andres.Vazquez-Torres@cuanschutz.edu

thiol-disulfide exchange reactions characteristic of the former because SsrB binds to thioredoxin variants lacking catalytic cysteine residues[22].

The genome of the human pathogen *Salmonella enterica* serovar Typhimurium encodes 30 different two-component regulatory systems, including OmpR and PhoP that activate transcription of *ssrB* gene within the SPI-2 gene cluster[18,23–25]. Because OmpR, PhoP and SsrB share the typical two-domain organization of response regulators[3] and because the 4-stranded β-sheets and 3-flanking α-helices are conserved in thioredoxin family members[26], we were intrigued by the possibility that several members of the response regulator family might have adopted thioredoxin as an allosteric regulator.

Here, we show that thioredoxin associates with structurally diverse members of the response regulator family. Our research identifies unique areas in thioredoxin and the response regulator OmpR that make such interactions possible. Structural, biochemical and functional approaches demonstrate that the widespread binding of response regulators to thioredoxin is vital for the transcriptional activation of SPI-2-dependent *Salmonella* virulence programs.

## Results

### Thioredoxin serves as a docking station for multiple regulatory proteins

To identify binding partners of *Salmonella*'s thioredoxin protein encoded by the *trxA* gene, we performed tandem-affinity purification (TAP) using a C-terminal fusion of thioredoxin with calmodulin-binding peptide and Protein A. Mass spectrometry (MS) analysis of proteins within the 15-60 kDa range identifies the response regulators OmpR and PhoP as TrxA-binding partners (Fig. 1a). Several other helix-turn-helix regulators, chaperones, metabolic and antioxidant proteins are also pulled down by the TrxA-TAP construct (Table S1, Fig. S1b), findings that are consistent with the chaperone properties of the thioredoxin protein of *E. coli*[27]. The binding of TrxA to FruR, GalS, GntR, OmpR, PdhR, PhoP, and YjfQ is verifiable in a bacterial two-hybrid system (Fig. 1b). With the exception of *galS*, a TrxA variant lacking the catalytic Cys[33] and Cys[36] residues interacts with all of these transcription factors as efficiently as wild-type thioredoxin ($p > 0.05$), suggesting that the binding of thioredoxin to these proteins does not involve thiol-disulfide exchange. Accordingly, the PhoP response

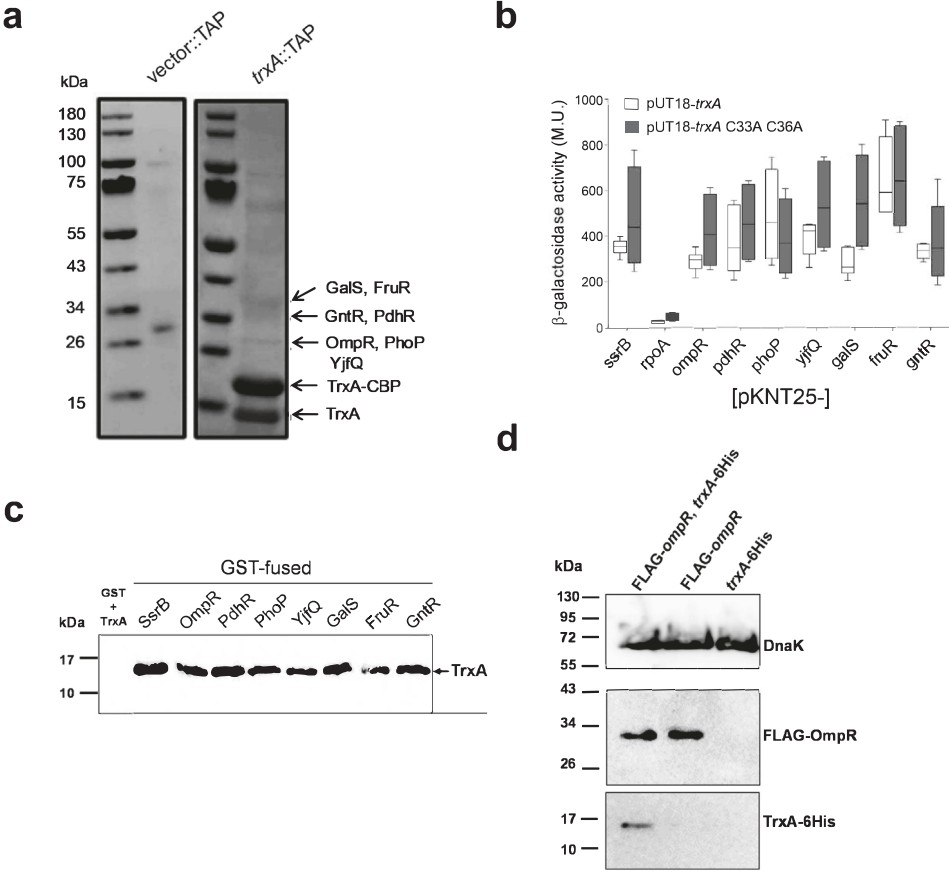

**Fig. 1 | Thioredoxin binds to multiple transcriptional regulators. a** Putative TrxA binding partners were identified by mass spectrometry of cytoplasmic proteins isolated by tandem affinity purification (TAP) from Δ*trxA Salmonella* harboring pWSK29::TAP (vector::TAP) or pWSK29::*trxA*::TAP (*trxA*::TAP). Soluble proteins from bacterial cells grown in LB broth to OD$_{600}$ of 3.2 were isolated by sonication in lysis buffer. Soluble proteins, which were affinity-purified with IgG Sepharose and calmodulin resins, were separated on 4-12% NuPAGE gradient gels in MES SDS running buffer and visualized by Imperial Coomassie Brilliant Blue staining. The positions of TrxA-binding partners identified by mass spectrometry are indicated by arrows. **b** Interactions of TrxA or TrxA C33A C36A variant with the identified transcription factors was assessed in a T18/T25 bacterial two-hybrid system. RpoA was used as a negative control. β-galactosidase activity was determined in samples collected at the stationary phase. The data are the mean ± SD (pUT18-*trxA*; *n* = 6, 6, 6, 8, 8, 6, 6, 6, pUT18-*trxA* C33A C36A; *n* = 8 biological replicates) from 2–3

independent experiments. With the exception of *galS*, no statistical differences ($p > 0.05$) were found in the interactions of target proteins with pUT118-trxA or pUT18 trxA C33A C36A as assessed by two-way ANOVA with Šidák multiple test correction. Whiskers in box plots represent minimal to maxima; 25th and 75th percentiles and median are also represented. **c** Recombinant TrxA-6His pulled-down by GST-tagged fusion recombinant proteins was detected by immunoblotting. GST was used as a negative control. The data are representative of 2–3 independent experiments. **d** Binding of TrxA::6His and FLAG::OmpR proteins expressed from their native chromosomal loci was visualized by immunoblotting of immunoprecipitates recovered from extracts of *Salmonella* grown overnight in LB broth. TrxA::6His proteins were immunoprecipitated with antibodies directed to FLAG-OmpR. The blots are representative of eight independent experiments. DnaK protein in the input lysates used for the immunoprecipitation was visualized by Western blotting. Source data are provided as a Source Data file.

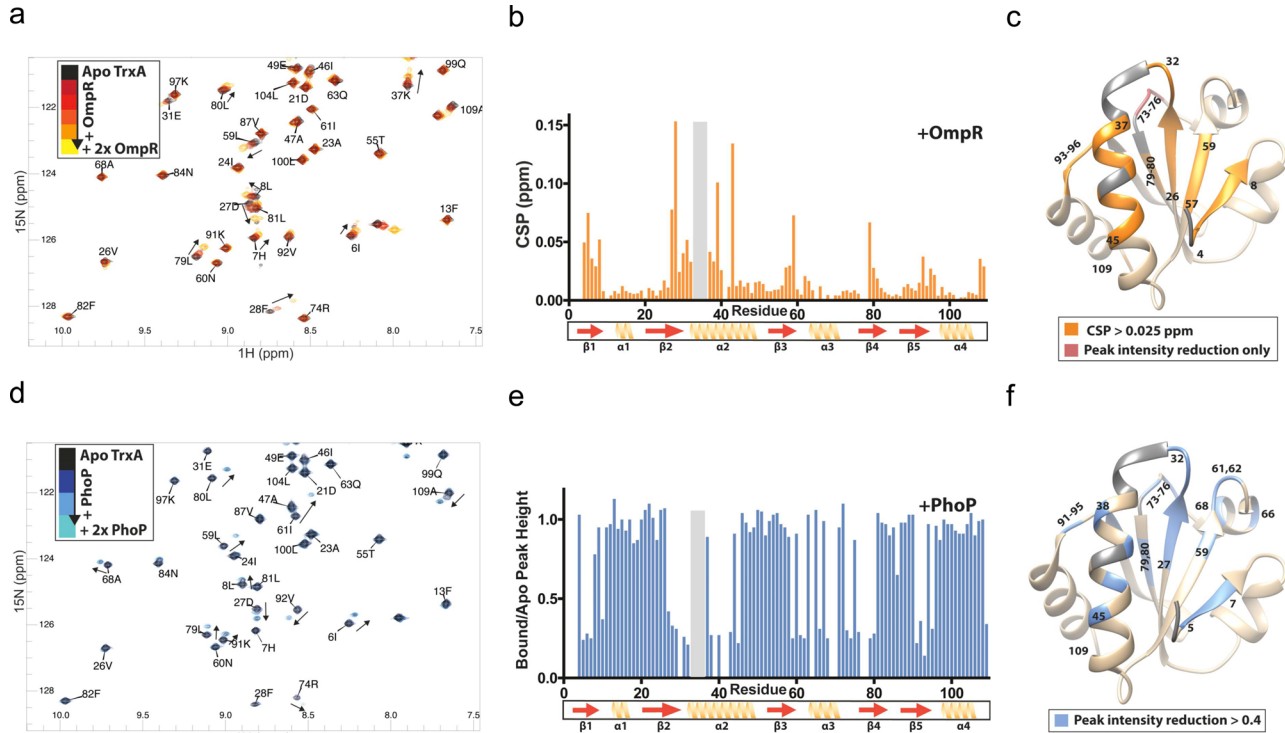

**Fig. 2 | The TrxA binding interface is shared between OmpR and PhoP. a** and **d** Overlay of ¹⁵N-HSQC spectra of apo TrxA (black) with increasing amounts of either unlabeled OmpR (yellow, **a**) or PhoP (blue, **d**). The ¹⁵N-HSQC spectra show peaks that correlate chemical shifts of covalently bonded ¹H and ¹⁵N atoms on the x- and y-axes, respectively. **b** Chemical shift differences between apo TrxA and TrxA:OmpR complexes (1:2 ratio), indicative of immediate chemical environment change, i.e., protein-protein interaction. **e** Relative peak intensity quenching in ¹⁵N-HSQC spectra of TrxA upon 1:2 titration with unlabeled PhoP, another indicator of immediate chemical environment change. Residues in the grey area could not be assigned in the spectra. TrxA residues responsible for interacting with OmpR **c** and PhoP (**f**) are plotted on the PDB crystal structure 4HUA. The residues of TrxA interacting with OmpR and PhoP are numbered. Grey residues in helix 2 are the missing amides from the HSQCs. Source data are provided as a Source Data file.

regulator, which lacks cysteine residues in its amino acid sequence, is among the clients of thioredoxin. TrxA can be pulled-down in a reconstituted biochemical system with the response regulators GST-SsrB, -OmpR, -PhoP, or -NarL, but not the GST protein control (Fig. 1c, S1c–e), demonstrating broad binding of thioredoxin to a variety of structurally diverse response regulators. GST-PdhR, -GalS, -FruR, -GntR, -YjfQ fusions also pull-down TrxA. Importantly, OmpR and TrxA expressed from their native loci in the chromosome interact in the cytoplasm of *Salmonella* (Fig. 1d), suggesting that the binding of this response regulator to thioredoxin occurs under biologically relevant conditions. As described for several members of the thioredoxin superfamily[27,28], our investigations indicate that thioredoxin is a broadly utilized chaperone in *Salmonella*. The catalytically deficient TrxA C33A C36A variant pulls down all regulatory factors tested, except GalS (Fig. S1f, S1g). Thus, in contrast to the holdase activity of other thioredoxin family members[27–29], *Salmonella*'s thioredoxin protein TrxA acts as a redox-independent chaperone, a situation reported for eukaryotic protein disulfide isomerase[30,31]. The widespread chaperone activity of thioredoxin identified herein provides a reasonable explanation for the remarkable role thioredoxin plays in *Salmonella* pathogenesis in the absence of its characteristic oxidoreductase activity[22].

**Mapping of thioredoxin interfacial residues that mediate binding to response regulators**

To identify the substrate recognition interface that mediates binding of the thioredoxin TrxA to response regulators, we used nuclear magnetic resonance (NMR) spectroscopy. NMR is uniquely suited for identifying site-specific interactions in solution via sensitive measurement of resonances (peaks in spectra) in various conditions. Upon ligand binding (i.e., direct interaction with the response regulators), residues in the protein-protein interface will undergo changes in the immediate chemical environment resulting in residue-specific resonance shifts and/or changes in peak intensity. We have assigned over 95% of the ¹⁵N,¹³C-labeled TrxA backbone resonances through conventional 3D NMR experiments. Notably, the region left unassigned in the spectra is the active site (residues 31-35), which shows exchange broadening likely due to its dynamic nature. According to secondary structure propensity (SSP) analysis based on chemical shifts. The α-helical and β-strand propensities of recombinant TrxA (Fig. S2Aa, S2b) match well to the previously solved thioredoxin crystal structure 4HUA[32]. Titration of unlabeled OmpR into ¹⁵N-labeled TrxA shows ¹⁵N-HSQC chemical shift perturbations (CSPs) in β-sheet and α-helix residues that lay near the catalytic site of TrxA (Figs. 2a–c, S2c). Distal C-terminal residues 108 and 109 also show CSPs, potentially due to an allosteric conformational change induced by OmpR binding. The regions with the largest decrease in peak intensity match closely with the regions that endure significant CSPs (Fig. S2d). Residues 74-76 undergo decreases in peak height upon OmpR treatment, suggesting that OmpR also interacts tightly with this region of TrxA. The stoichiometric binding behavior of the titration indicates that the binding occurs in the intermediate exchange regime on μs-ms time scale with a $K_d$ under 10 μM.

We have tested whether TrxA's substrate recognition motif is shared among response regulators and have repeated HSQC titrations of ¹⁵N-TrxA with recombinant PhoP (Figs. 2d, S2a, S2e). Compared to OmpR, where we mainly see CSPs typical of fast binding exchange, the interaction between TrxA and PhoP occurs with likely higher affinity due to the presence of two separate sets of peaks for bound and unbound TrxA that are characteristic of slow exchange (>1 ms). Since

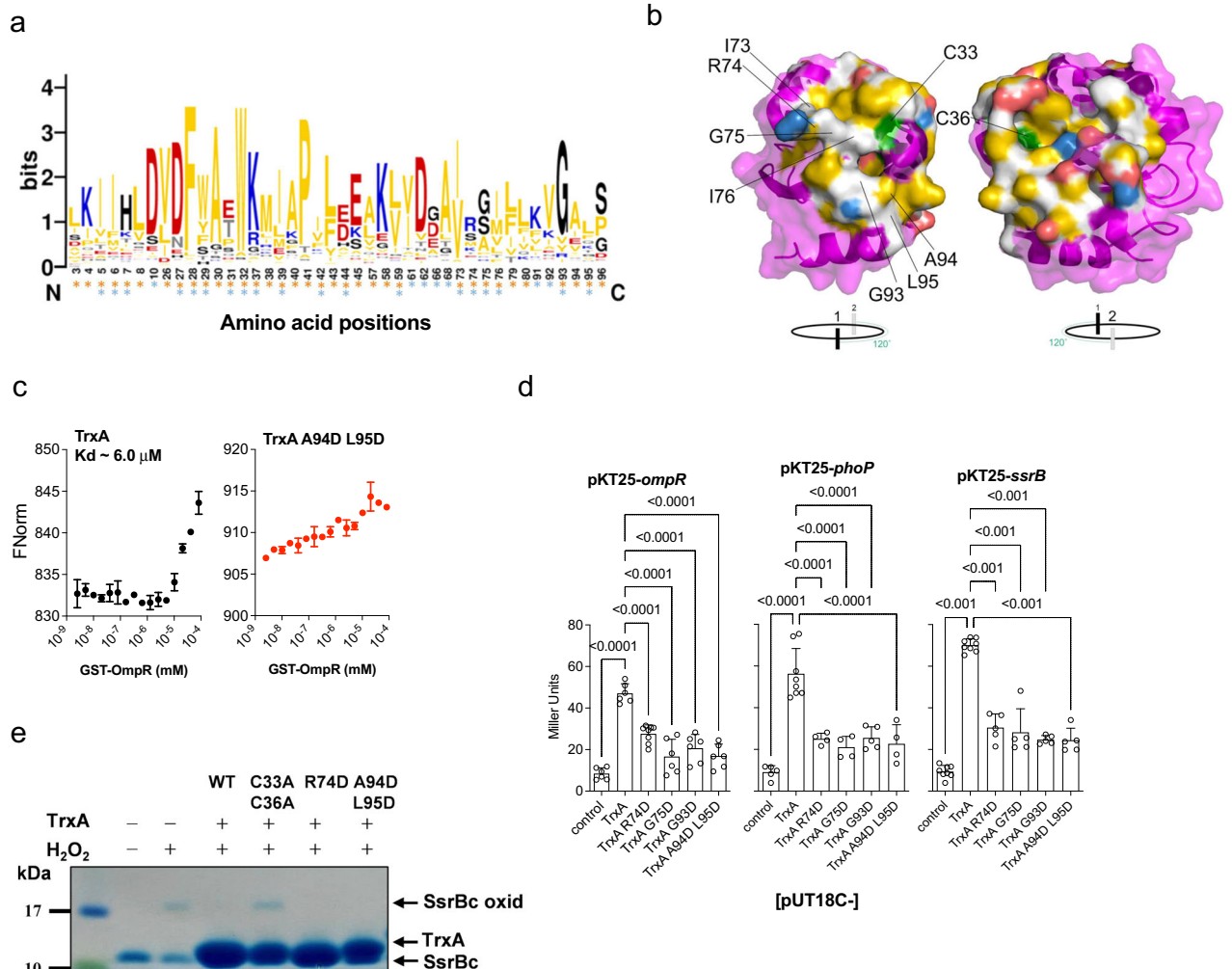

**Fig. 3 | The TrxA interface serves as docking station to response regulators.**
**a** A consensus logo generated from results of a 3D-BLAST search on TrxA (PDB 4HUA) after alignment by ClustalW, n = 448, e-value < 1⁻¹⁰. Asterisks represent thioredoxin-binding residues that interact with OmpR (orange) or PhoP (blue).
**b** The ∼ 4400 Å² TrxA binding interface that mediates associations to OmpR or PhoP is highlighted blue, red, yellow, or gray to show positive, negative, hydrophobic, or hydrophilic identities, while the catalytic cysteine residues are in green. Residues probed by mutagenesis are labeled. **c** Binding affinity of the wildtype TrxA or the TrxA A94D L95D variant to GST-OmpR was estimated by microscale thermophoresis. The calculated Kd value is displayed. The data are mean ± SD from 2 independent experiments. **d** Interaction of TrxA variants with OmpR, PhoP, and SsrB as assessed in a bacterial two-hybrid system. *E. coli* strains expressing pUT18C empty vector with pKT25-*ompR*, -*phoP*, or -*ssrB* were used as a negative control. β-

galactosidase activity was measured in *E. coli* cells grown to stationary phase in LB broth. The data are mean ± SD (pKT25-*ompR*; n = 6, pKT25-*phoP*; n = 5, 8, 4, 4, 5, 5, pKT25-*ssrB*; n = 8, 8, 5, 5, 5, 5 biological replicates) from 3-5 independent experiments. ***p < 0.001 as evaluated by one-way ANOVA with Dunnett's multiple test correction. df = 5; F = 34,32;101,2; and 26,18 for *ompR*, *phoP*, and *ssrB*, respectively. **e** The in vitro oxidoreductase activity of TrxA variants towards a disulfide-bonded C-terminal fragment of SsrB (SsrBc) containing the DNA-binding and dimerization domains. Disulfide-bonded SsrBc proteins were obtained after treatment with 500 μM H₂O₂ for 1 h at 37 °C. Where indicated, TrxA proteins were added to the reactions. Specimens were separated on non-reducing gradient SDS-PAGE gels, and the proteins in the gels were stained with Imperial Coomassie Brilliant Blue. Data are representative of five independent experiments. Source data are provided as a Source Data file.

the peaks in the interfacial area cannot be fitted using CSPs, we have measured the change in the apo peak intensity of TrxA upon PhoP addition (Fig. 2e). This approach reveals that TrxA residues sustaining a large reduction in peak intensity upon the addition of PhoP remarkably overlap with those experiencing notable CSPs following OmpR infusion. These data demonstrate conservation in the interfacial surface that couples thioredoxin to OmpR and PhoP (Fig. 2c, f). In addition to the shared binding residues, OmpR and PhoP also sustain specific interactions with thioredoxin. For example, residues 61-68 in α-helix 3 are perturbed only upon PhoP binding.

### Thioredoxin interfacial residues mediate binding to response regulators

We next experimentally tested whether thioredoxin interfacial residues identified by NMR do indeed mediate interactions with several

response regulators. Nonpolar and hydrophobic residues predominate at the center of the TrxA substrate binding area (Fig. 3a, b), whereas charged residues, such as the highly conserved Arg⁷⁴ that is located in the flexible loop connecting α3 and β4, are enriched at the rim (Figs. 3b, S3a, S3b). The NMR analysis indicates that OmpR and PhoP share binding to thioredoxin's Ile⁷³, Arg⁷⁴, Gly⁷⁵, Gly⁹³, Ala⁹⁴ and Leu⁹⁵ conserved residues (Fig. 3a). To test the contribution of the latter residues to the binding of thioredoxin to response regulators, we have purified recombinant TrxA proteins bearing aspartic acid substitutions in the predicted binding residues (Fig. S4a). Aspartic acid has been chosen for the mutagenesis because the interfacial surface is quite extensive, occupying about 4,400 Å². Aspartic acid-bearing TrxA variants preserve wild-type secondary structure (Fig. S4b), and are expressed by *Salmonella* at similar or higher levels than wild-type thioredoxin (Fig. S4c, S4d). In agreement with the intermediate

exchange character of the NMR titrations, microscale thermophoresis (MST) shows that thioredoxin binds to OmpR with an estimated $K_d$ of 6 µM (Fig. 3c), a binding constant that is in agreement with the estimated concentration of OmpR in the cytoplasm of *E. coli*[33]. The binding of OmpR to recombinant TrxA R74D, G75D, I73D I76D, G93D, or A94D L95D variants bearing mutations at the predicted substrate binding motif is below the detection limit of MST (Figs. 3c, S4e), demonstrating that Ile[73], Arg[74], Gly[75], Ile[76], Gly[93], Ala[94] and Leu[95] actively participate in the binding of thioredoxin to OmpR. TrxA variants bearing I6D or A88D aspartic acid substitutions have normal secondary structure and bind to recombinant OmpR protein with a $K_d$ of 28 and 36 µM, respectively (Fig. S4a, S4b, S4e), suggesting that not all aspartic acid substitutions destroy binding of TrxA to OmpR. According to this idea, TrxA proteins bearing the catalytic site mutations C33D and C36D also appear to bind normally to recombinant OmpR protein (Fig. S4f, S4g). Moreover, TrxA variants bearing alanine mutations in the catalytic Cys[33] and Cys[36] residues bind to OmpR with a $K_d$ value of 16 µM (Fig. S4e), in agreement with NMR CSP that shows minor CSP or peak quenching upon binding.

A bacterial two-hybrid system confirms that thioredoxin binds to OmpR, PhoP and SsrB (Fig. 3d). As recorded for the OmpR protein, Arg[74], Gly[75], Gly[93], Ala[94] and Leu[95] are also important for binding of thioredoxin to the response regulators SsrB and PhoP (Fig. 3d), demonstrating that the OmpR-defined thioredoxin binding surface serves as a docking station for multiple response regulators.

Thioredoxin is one of the most prevalent antioxidant defenses of eukaryotic and prokaryotic cells alike[34]. The catalytic cysteines mediate thiol disulfide exchange reactions by funneling NADPH reducing power through a flavin cofactor. Because the thioredoxin interfacial surface identified in our investigations lays in the vicinity of the catalytic cysteine residues, we deemed it important to ascertain whether the aspartic acid mutations engineered in the binding area affect oxidoreductase enzymatic activity. As expected, the control TrxA C33A C36A catalytic mutant lacks oxidoreductase activity (Fig. 3e). In contrast, R74D and A94D L95D TrxA variants are fully capable of reducing disulfide-bonded SsrB (Fig. 3e). Collectively, these findings demonstrate that the thioredoxin substrate-binding surface uncovered in our investigations mediates chaperone activity for diverse response regulators, but is dispensable for canonical oxidoreductase activity.

### Thioredoxin interfacial residues enable transcription of *Salmonella* virulence genes

Thioredoxin is essential for *Salmonella* virulence[22,35,36]. However, *Salmonella* do not rely on the canonical oxidoreductase activity of thioredoxin for growth in macrophages or mice[22]. Our identification of mutant alleles that specifically destroy interaction of TrxA with various clients without affecting canonical oxidoreductase activity enable us to test the specific contribution of thioredoxin's chaperone activity to *Salmonella* pathogenesis. Mutations of TrxA interfacial surface residues abate growth of *Salmonella* in J774 cells (Fig. 4a), but do not affect their growth in LB broth (Fig. S5a). Aspartic acid substitutions do not indiscriminately impact the ability of *Salmonella* to grow intracellularly as demonstrated by comparable growth between wild-type *Salmonella* and *Salmonella* strains bearing *trxA* A16D, K19D, C33D, C36D, A68D, A73D, A80D, K83D, V87D, A88D, A89D, V92D, L95D, G98D, A106D, L108D, and A109D mutant alleles in J774 cells (Fig. S5b, S5c). Because thioredoxin interfacial residues enable contact of this chaperone with OmpR, PhoP and SsrB, which are known to activate transcription of SPI-2 genes[21,24,25,37], we measured transcription of the SPI-2 genes *ssaV* and *sifA* in intracellular *Salmonella* bearing wild-type or mutant *trxA* alleles. Our research demonstrates that thioredoxin's interfacial residues facilitate transcription of *ssaV* and *sifA* genes encoding a structural component and effector of the SPI-2 type III secretion system, respectively, as well as PhoP- and OmpR-regulated *mgtA* and *ompF*

genes, by intracellular *Salmonella* (Fig. 4b). Moreover, *Salmonella* expressing thioredoxin interfacial mutations are at a competitive disadvantage when coinoculated intraperitoneally into C57BL/6 mice with equal numbers of wild-type *Salmonella* (Fig. 4c). Cumulatively, our investigations demonstrate that the chaperone activity of thioredoxin is critical for both the transcription of *Salmonella* pathogenicity island-2 genes and for the virulence of this intracellular pathogen in macrophages and mice.

### Mapping of OmpR interfacial residues that make contact with thioredoxin

Having identified and functionally characterized the importance of thioredoxin interfacial residues in binding to response regulators and their transcriptional outputs, we used NMR to identify the amino acids that couple response regulators to thioredoxin. Few full-length structures for C- and N-terminal domains of response regulators have been solved, but none exist for OmpR, PhoP, or SsrB. We focused our efforts on the archetypical response regulator OmpR, from which much information has been learned about this class of signaling molecules[1,4,38,39]. To utilize 3D Transverse-Relaxation Optimized Spectroscopy (TROSY) pulse-sequences in combination with the deuteration of non-exchangeable protons, we have produced high concentrations of soluble OmpR protein (Fig. S6a) amenable to NMR analysis (Fig. S6b). We have been able to assign ~60% of the OmpR backbone, missing amino acids 50-105 in the N-terminal receiver domain that are presumably highly dynamic residues. The chemical shift-derived secondary structure propensity agrees well with the structure predicted using Phyre2, which confirms the two-domain composition of this response regulator (Fig. S6c). The intradomain structures predicted by AlphaFold match similarly solved crystal structures of the receiver domain of NarL (pdb:1RNL[40]) and the C-terminal domain of OmpR (pdb:1OPC[39]). Titration of unlabeled thioredoxin into [15]N-labeled OmpR results in shifts in Ala[136] of the unstructured linker, Ile[138] of the nearby $\beta_6$-sheet as well as Arg[182] and Tyr[230] of the C-terminal wing domain of OmpR (Figs. 5a, b, S6b). This biophysical approach shows that the OmpR linker and the C-terminal domain are footprinted when in contact with thioredoxin.

### OmpR interfacial mutants bind poorly to thioredoxin

Using a variety of reconstituted biochemical and genetic platforms, we have tested the degree that OmpR interfacial residues identified by NMR contribute to the binding of this response regulator to thioredoxin. In agreement with the NMR data, OmpR variants expressing the A136D or I138D alleles exhibit poor binding to TrxA in pull-down assays (Figs. 6a, S7a, S7b). Moreover, OmpR R182D and Y230D mutations mapping to dimerization and β-hairpin DNA-binding regions[38], respectively, associate with lower affinity to TrxA than wild-type OmpR (Figs. 6a, S7a, S7b). The importance of Ala[136], Ile[138], Arg[182] and Tyr[230] in binding of OmpR to TrxA is confirmed in a bacterial two-hybrid system (Fig. 6b). The poor binding of these OmpR variants to thioredoxin does not appear to be due to misfolding (Fig. S7c), nor inability to interact with the cognate sensor kinase EnvZ. In fact, OmpR A136D, OmpR I138D, OmpR R182D, or OmpR Y230D are phosphorylated by a C-terminal cytoplasmic fragment of the sensor kinase EnvZ as effectively as the wildtype OmpR protein (Figs. 6c, S7d, S7e, S7f). Moreover, OmpR variants expressing the A136D, I138D, R182D or Y230D mutations bind with apparently wild-type affinity to the α subunit RpoA of RNA polymerase (Figs. 6d, S7g). These variants also associate to a DNA fragment containing the *ompF* promoter (Figs. 6e, S7h). Except for OmpR R182D, all other mutants activate similar levels of *ompF* in vitro transcription when compared to the wild-type protein (Fig. 6f). As noted previously[41,42], an OmpR variant bearing an alanine substitution in place of the phosphorylatable aspartic acid in

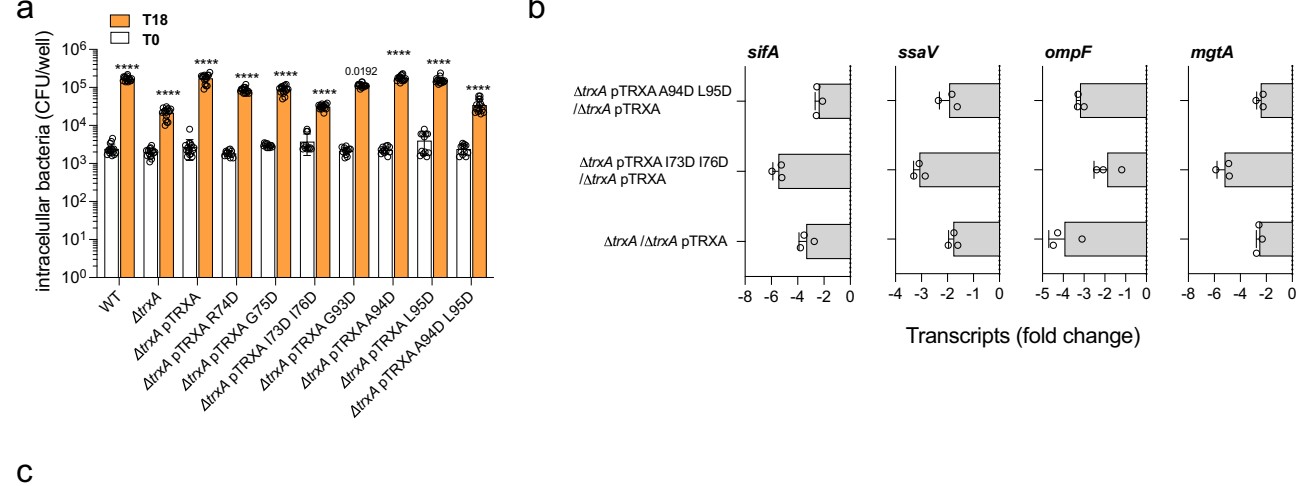

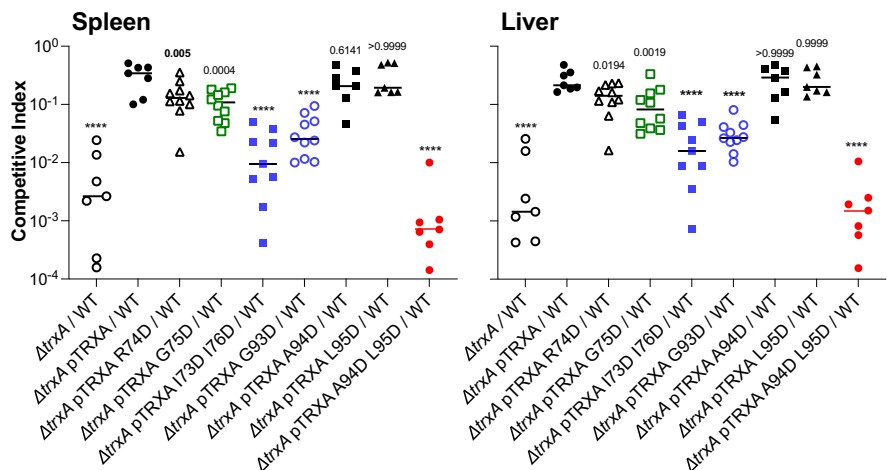

**Fig. 4 | Residues in the shared TrxA substrate-binding interface contribute to *Salmonella* pathogenesis. a** Intracellular replication of the indicated *Salmonella* strains in J774 cells was evaluated by recording CFU at times 0 and 18 h postinfection. The data are the mean ± SD (T0; *n* = 16, 16, 16, 12, 12, 12, 12, 12, 12, 12, T18; *n* = 16 biological replicates). ****p < 0.0001, ***p < 0.001 as estimated by two-way ANOVA with Dunnet's multiple test correction. **b** The transcription of SPI-2 (*sifA* and *ssaV*), *ompF*, and *mgtA* genes 8 h after infection of J774 cells with the indicated *Salmonella* strains was estimated by qRT-PCR following the threshold cycle ($2^{-\Delta\Delta CT}$) method. The data are the mean ± SD (*n* = 3 biological replicates) from three independent experiments. Gene expression was normalized to internal levels of the housekeeping gene *rpoD*. Transcripts downregulated ≥ 2-fold were considered significant. **c** Bacterial burden in spleens and livers of C57BL/6 mice 96 h after i.p. inoculation with ~1000 CFU of a mixture containing equal numbers of wild-type *Salmonella* and isogenic strains expressing *trxA* interfacial mutations. Horizontal bars represent the median from 7–10 mice collected in 2 independent experiments (*n* = 7, 7, 10, 10, 9, 10, 7, 7, 7). Exact p values are shown, and *p* < 0.0001 as calculated by one-way ANOVA when compared to Δ*trxA* pTRXA/WT. df = 8; F = 1375 and 1429 for spleen and liver, respectively. Source data are provided as a Source Data file.

the N-terminal domain binds poorly to the *ompF* promoter (Figs. 6e, S7h), and does not activate *ompF* in vitro transcription (Figs. 6f, S7h). Aspartic acid substitutions of Ala[136], Ile[138] or Tyr[230] yield insoluble OmpR proteins in the cytoplasm of *Salmonella* (Fig. 6g). As described for SsrB[22], wildtype OmpR segregates in the insoluble fraction in Δ*trxA Salmonella* (Fig. S7i). In contrast, the OmpR E178K variant, which binds to thioredoxin with seemingly greater affinity than wild-type OmpR, partions in the soluble fraction (Fig. S7j). Together, our investigations have identified an extensive region in OmpR, which predominantly encompasses residues in the flexible linker and C-terminal domain that mediates contact with thioredoxin.

### OmpR interfacial residues facilitate the transcriptional activation of *Salmonella* virulence programs

*Salmonella* relies on the OmpR response regulator to grow intracellularly (Fig. 7a). The *A136D, I138D,* or *Y230D* mutations in OmpR's interfacial residues located at the flexible linker or the structured C-terminal domain diminish intracellular growth of *Salmonella* in macrophages (Fig. 7a), but do not compromise growth in LB broth

(Fig. S7k). In addition to participating in the interfacial surface that mediates binding to TrxA (our data above), Arg[182] is involved in dimerization[38]. This might contribute to the stronger intracellular growth defect of *ompR R182D Salmonella* compared to controls expressing the *ompR A136D, I138D,* or *Y230D* alleles. For this reason, we have chosen *Salmonella* strains bearing OmpR A136D, I138D or Y230D mutations for further testing. OmpR has been incorporated into the circuitry that activates the transcription of the *Salmonella ssrA* gene[24]. *Salmonella* strains expressing the A136D, I138D, or Y230D mutations in OmpR interfacial residues express lower levels of the *ssrA* gene 8 h after infection of J774 cells (Fig. 7b). *Salmonella* strains bearing mutations in *ompR* interfacial residues also express lower levels of the *sifA*-encoded SPI-2 effector (Fig. 7b). *Salmonella* expressing OmpR A136D, I138D or Y230D variants, which do not bind to thioredoxin but associate with RNA polymerase as well as cognate sensor kinase and DNA, are attenuated in an acute model of infection (Fig. 7c). Our investigations demonstrate a critical role for the interactions of OmpR with thioredoxin in *Salmonella* pathogenesis. The role played by OmpR interfacial residues in *Salmonella* pathogenesis can be explained by the contribution that these residues play in the global

a

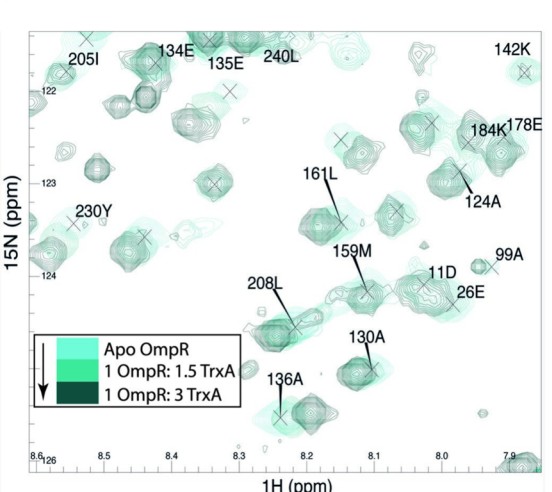

b

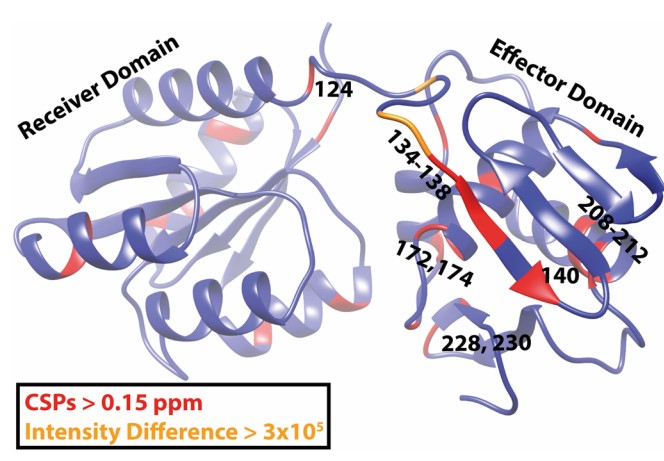

**Fig. 5 | Identification of residues in OmpR that arbitrate interactions with TrxA. a** Region of [15]N-HSQC spectra of apo OmpR (cyan) with increasing amounts of TrxA (dark green). **b** Major CSPs and intensity changes from the [15]N-HSQC spectra are plotted onto the predicted structure of OmpR. Source data are provided as a Source Data file.

regulation of *Salmonella* genes, including the horizontally-acquired SPI-2 gene cluster that encodes a vital type III secretion system[17,24,37] as well as classical OmpR-regulated targets.

## Discussion

Two-component signaling systems control a multitude of virulence programs and countless physiological processes in bacteria, archaea and eukaryotes. Our investigations demonstrate that the degenerate affinity of response regulators for thioredoxin is vital for the faithful transmission of two-component signaling required for *Salmonella* pathogenesis. Congruent binding of response regulators onto the thioredoxin surface area discovered in our investigations is a pre-requisite for the transcriptional expression of a cluster of genes that encodes a type III secretion system essential for *Salmonella* pathogenesis. Recombinant OmpR proteins are deficient in thioredoxin binding associate with cognate sensor kinase and promoter regions to activate transcription in a reconstituted biochemical system. However, *Salmonella* expressing OmpR variants with poor binding capacity for thioredoxin are attenuated in mice, grow poorly in macrophages, and are unable to activate normal levels of SPI-2 genes intracellularly. To properly transmit virulence programs in the crowded environment of the cytoplasm, response regulators such as OmpR require the chaperone activity provided by thioredoxin. The insolubilty in *Salmonella* of OmpR A136D, I138D, or Y230D variants, which exhibit poor binding with thioredoxin, resembles the segregation of SsrB to inclusion bodies in Δ*trxA Salmonella*[22]. OmpR variants are, however, soluble as recombinant proteins at dilute conditions, suggesting that TrxA helps with solubilization of response regulators in the crowded environment of bacterial cytoplasms. Keeping response regulators in a functional, soluble fraction in *Salmonella*'s cytoplasm could contribute to the optimal transcription of virulence genes, although other models by which the chaperone activity of thioredoxin influences the signaling properties of response regulators cannot be ruled out at present. Our investigations demonstrate that the interactions that OmpR establishes with thioredoxin are a critical aspect of the signal transduction pathway that activates *Salmonella* virulence.

The specificity inherent to sensor kinases and their cognate response regulators guarantees reliable transmission of signals within the many two-component pairs that coexist in a single cell[4–7]. Our research with NarL, OmpR, PhoP and SsrB demonstrates that the astonishing specificity between response regulators and sensor kinases sharply contrasts with the promiscuity with which response

regulators partner with thioredoxin. Signaling through two-component systems balances the highly discriminatory partnering between histidine kinases and their cognate response regulators with the promiscuous binding of response regulators to thioredoxin. Associations of response regulators to thioredoxin involves expansive areas in both proteins, indicating that physicochemical traits, rather than specific interactions, enable the binding of structurally related proteins to thioredoxin. The pull-down assays did not identify all response regulators encoded in the *Salmonella* genome as partners of thioredoxin, which could reflect the specificity of this chaperone for some response regulators but not others. However, the TAP screen in our investigations has some limitations. For example, SsrB and NarL were not identified in our TAP assays, likely because these response regulators are expressed following nutritional limitations and acidic pH or anaerobic nitrate respiration, respectively. Nonetheless, recombinant NarL and SsrB proteins bind to TrxA. More work will be needed to determine the extent of the interaction between response regulators and thioredoxin.

Other than the transmission of input and output signals between the modular domains of response regulators, no specific function is known for the flexible linker. Our investigations demonstrate that the flexible linker of OmpR mediates binding to thioredoxin. In addition to the linker, residues in the C-terminal domain of OmpR are footprinted by thioredoxin. Because the modular organization of response regulators is evolutionarily preserved, our findings are likely applicable to other members of this prevalent family of signaling molecules.

Few response regulators have been tracked structurally. We provide the first structure of the two domains of the prototypical response regulator OmpR, from which much knowledge of bacterial signaling has been derived. Our structural, biochemical, genetic, and functional approaches identify unique surface residues in OmpR that mediate binding to thioredoxin but are dispensable for the interactions of this transcription factor to either RNA polymerase or DNA. Our investigations also demonstrate that mutations of the interfacial residues in OmpR do not affect interactions of this response regulator with its sensor kinase EnvZ. In fact, the excellent binding of OmpR interfacial mutants to the *ompF* promoter and the ability of these variants to activate *ompF* in vitro transcription sharply contrast with the inability of the OmpR D55A mutant lacking the phosphorylatable aspartate to do either. Mutations in the interfacial residues Ala[136], Iso[138] or Tyr[230] have uncovered critical roles for the interactions of OmpR with the chaperone TrxA in *Salmonella* pathogenesis.

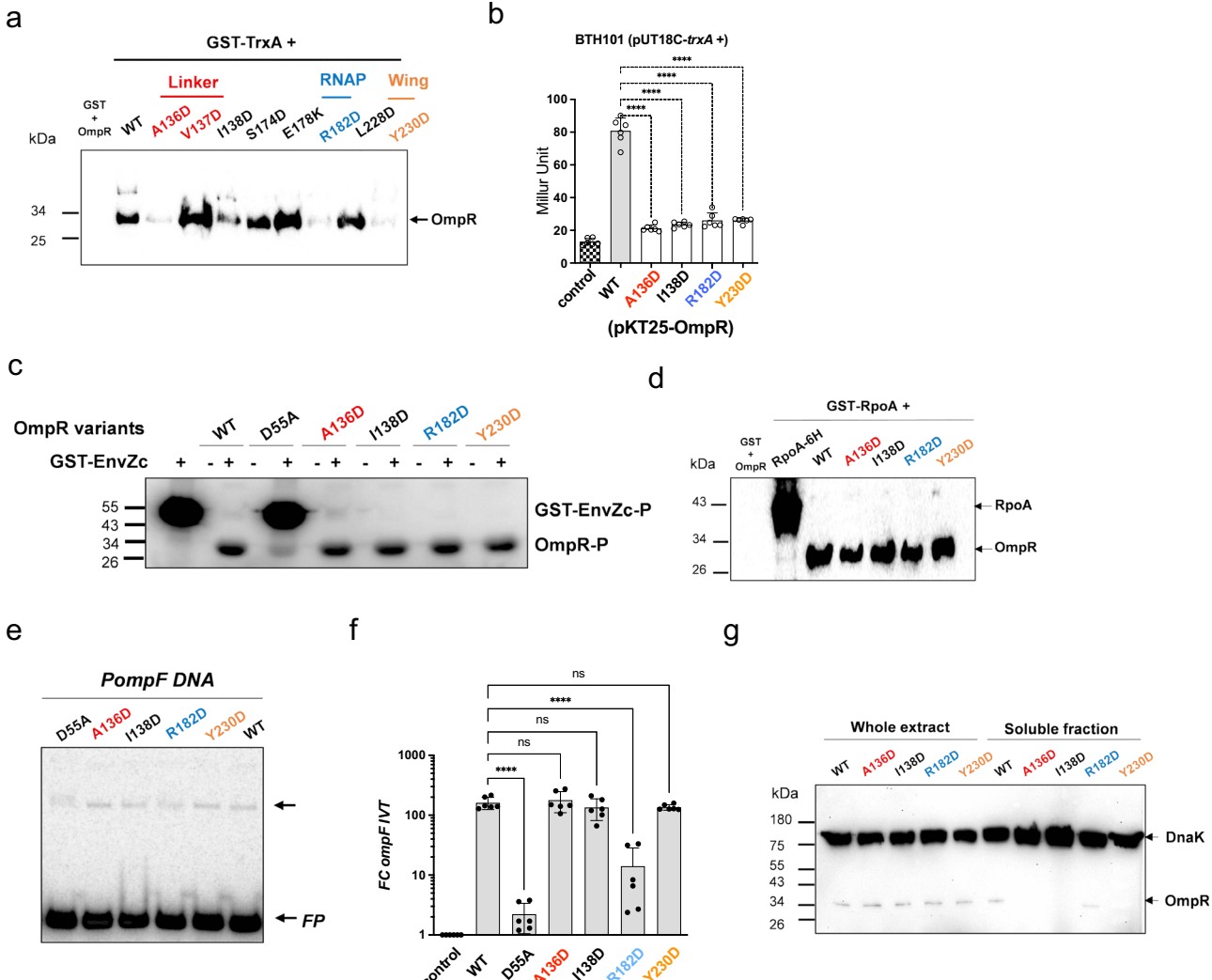

**Fig. 6 | Functional characterization of the OmpR interfacial residues.**
**a** Interaction of TrxA with OmpR variants bearing mutations in interfacial residues was evaluated in biochemical pull-down assays. Recombinant OmpR-6His variants pulled-down by GST-TrxA recombinant proteins were detected by immunoblotting using anti-His antibodies. GST was used as a negative control. The data are representative from 2–3 independent experiments. **b** In vivo binding of OmpR variants to TrxA was evaluated in a bacterial two-hybrid system in *E. coli* grown overnight in LB broth. The data are the mean ± SD ($n = 6$ biological replicates) from 3 independent experiments. ****$p < 0.0001$ as calculated by one-way ANOVA with Tukey's's multiple comparison test. df = 5, F = 224,3. **c** [γ-$^{32}$P]-ATP-dependent phosphorylation of OmpR variants by a fragment of encompassing 223-450 residues of the C-terminal cytoplasmic domain of EnvZ produced as a fusion protein with GST (i.e., GST-EnvZc) after 60 min of incubation at room temperature. Phosphorylated OmpR and EnvZc proteins were visualized by autoradiography. The gel is representative from 3 independent experiments. **d** Binding of FLAG-OmpR variants to the GST-RpoA α-subunit of RNA polymerase was analyzed in pull-down assays. The proteins were

probed by immunoblotting using anti-His antibodies. The blot is representative of three independent experiments. **e** Binding of OmpR variants to a DNA fragment containing the F2 and F3 OmpR binding site (*PompF*) was assessed by EMSA. The data are presentative from four independent experiments. The lower arrow labeled (FP) indicates the free probe, and the top arrow shows the complex of *PompF* with OmpR variants. **f** Activation of *ompF* in vitro transcription by OmpR proteins. The abundance of *ompF* transcripts was quantified by qRT-PCR. The data are mean ± SD ($n = 6$ biological replicates) from 3 independent experiments. ****$p < 0.0001$ as calculated by one-way ANOVA with Dunnet's multiple test correction. df = 6, F = 28,82. **g** Solubility of wild-type and OmpR variants expressing an N-terminus FLAG-tag was determined by immunoblot using antibodies to the FLAG-tag. FLAG-*ompR*-expressing *Salmonella* strains were grown in LB broth overnight at 37 °C. Immunoblotting of the DnaK protein was used as internal loading control. The blot is representative of three independent experiments. Source data are provided as a Source Data file.

Our structural and biochemical approaches have identified Arg$^{182}$ and Tyr$^{230}$ in the C-terminal effector domain as critical residues that mediate contact of OmpR with thioredoxin. Mutations in Arg$^{182}$ or Tyr$^{230}$ negatively impact the transcription of SPI-2 genes and attenuate *Salmonella* virulence. The latter phenotypes of OmpR variants bearing mutations in Arg$^{182}$ could reflect the previously identified function this charged residue plays in homodimer formation[38] and the herein-described role Arg$^{182}$ plays in binding to thioredoxin. Tyr$^{230}$ in the β-hairpin has previously been shown to mediate binding to cognate DNA. The sidechain of Tyr$^{230}$ not only mediates interactions of OmpR with backbone phosphate groups in the major groove of cognate DNA but

also establishes two H-bonds with Leu$^{180}$ in the homodimer[38]. We are surprised by the excellent DNA-binding and transcriptional activity supported by the OmpR Y230D variant. The greater contribution of nearby residues such as Thr$^{224}$, Trp$^{226}$, and Gly$^{227}$ to the binding of OmpR to cognate DNA[38] may compensate for the loss of apparently weaker contributions by Tyr$^{230}$. Nonetheless, a *Salmonella* strain bearing the OmpR Y230D variant transcribes SPI-2 genes poorly and is attenuated in macrophages and mice. Together, these findings suggest that the binding of OmpR to thioredoxin is a critical step in the transmission of signaling outputs that regulate critical aspects of *Salmonella* pathogenesis.

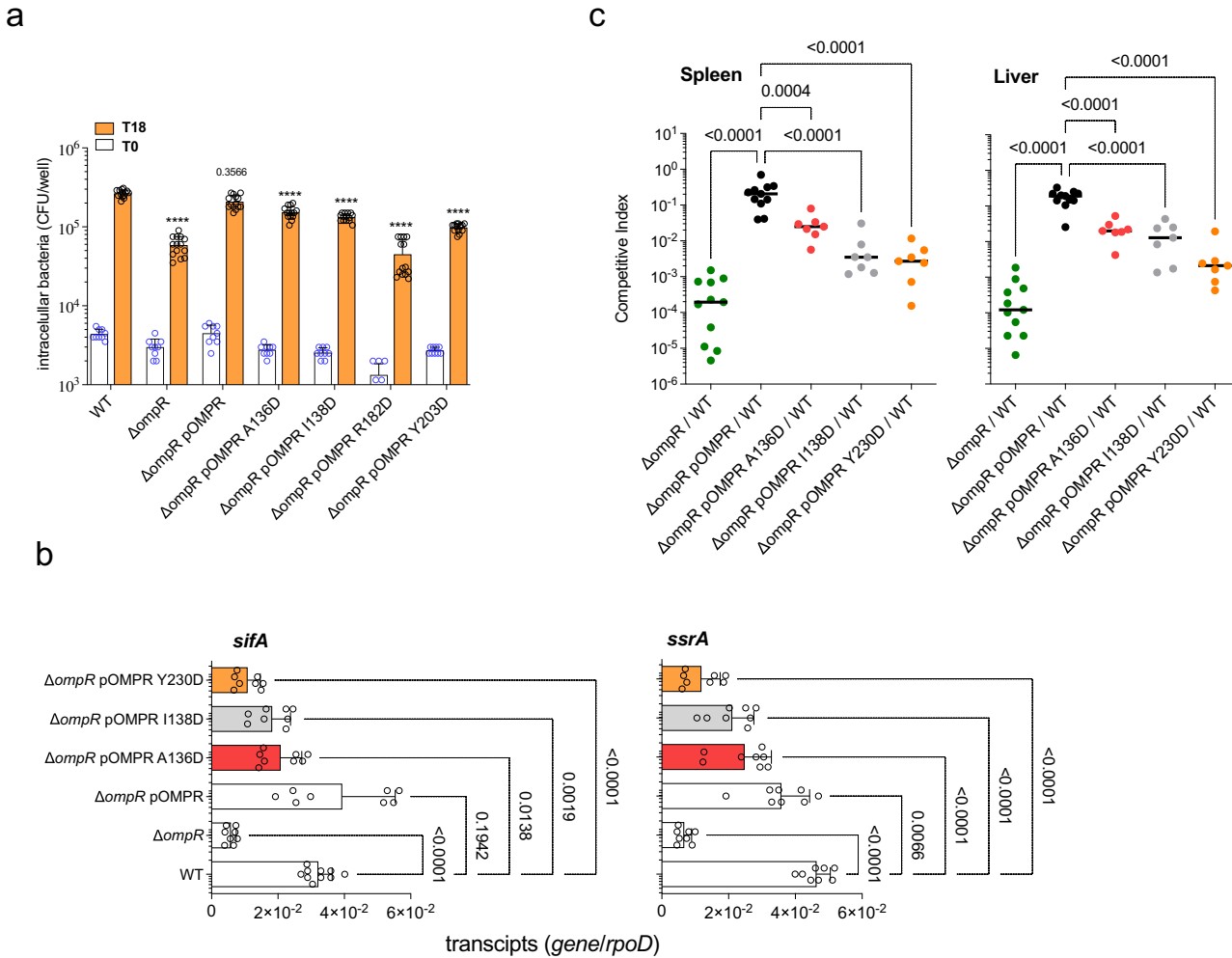

**Fig. 7 | OmpR interfacial residues mediating binding to TrxA contribute to *Salmonella* pathogenesis. a** Intracellular replication of the indicated *Salmonella* strains was evaluated by recording CFU at times 0 and 18 h after infection of J774 cells at an MOI of 2. The data are mean ± SD (T0; *n* = 9, T18; *n* = 14 biological replicates). ****$p < 0.0001$ as estimated by two-way ANOVA with Sidák's multiple test correction. **b** The abundance of SPI-2 transcripts in RNA specimens isolated from J774 cells 8 h after infection with *Salmonella* at an MOI of 50 was quantified by qRT-PCR. The results are normalized to internal levels of the housekeeping gene *rpoD*. The data are the mean ± SD (*sifA*, *n* = 10, 8, 8, 8, 8; *ssrA*, *n* = 8). Exact *p* values are shown and $p < 0.0001$ as calculated by one-way ANOVA when compared to wildtype controls. with Dunnet's multiple test correction. df = 5, F = 22,88. **c** Virulence of *Salmonella* bearing the indicated *ompR* alleles was assessed by recording competition in spleens and livers of C57BL/6 mice 96 h after i.p. inoculation of a ~1000 CFU mixture of equal numbers of wild-type and isogenic *ompR* mutant *Salmonella*. Horizontal bars are the median from 7–11 mice in 2 independent experiments (*n* = 11, 11, 7, 7, 7). ****$p < 0.0001$, ***$p < 0.001$ as evaluated by one-way ANOVA. df = 4, and F = 11,7 or 33,92 for spleen and liver, respectively. Source data are provided as a Source Data file.

Several of thioredoxin's conserved interfacial residues identified herein facilitate the binding of thioredoxin to functionally and structurally diverse substrates such as thioredoxin reductase or bacteriophage T7 DNA polymerase[43,44]. Arg[74] protruding from the surface of thioredoxin and the hydrophobic patch that includes Ile[76] are shared in the binding of thioredoxin to thioredoxin reductase[44] and response regulators (our investigations). Thus, the extensive surface area uncovered in our investigations likely mediates the binding of thioredoxin to diverse clients. The interfacial residues in TrxA's binding surface are generally shared between OmpR and PhoP; however, some residues in TrxA are uniquely engaged by either OmpR or PhoP. Other substrates may uniquely footprint some but not all interfacial residues as well, thereby providing a rationale for why not all mutations in all TrxA interfacial residues have the same impact in *Salmonella* virulence. Our research has made ascertaining chaperone and oxidoreductase enzymatic activities of thioredoxin possible. The functional properties of substrate-binding residues are phylogenetically conserved in members of the thioredoxin superfamily. It will be interesting to know whether canonical oxidoreductase or the less studied chaperone functions of thioredoxin evolved first.

## Methods

The research presented here was approved by the Institutional Biosafety Committee at the University of Colorado School of Medicine.

### Bacterial strains, plasmids, and growth conditions

The derivatives of *Salmonella enterica* serovar Typhimurium strain 14028s and *Escherichia coli*, as well as the plasmids used in this study, are listed in Tables S2 and S3. Deletion mutants were constructed using the λ-Red homologous recombination system[45]. Pfu ultra high-fidelity DNA polymerase (cat. # 600382-51, Agilent, Santa Clara, CA) was used to perform genetic mutagenesis using primers described in Table S4. Bacteria were grown in LB broth at 37 °C in a shaker incubator. Penicillin (cat. # J63032, AlfaAesar), chloramphenicol (cat. # 0230, Amresco), tetracycline (cat. # 0422, Amresco), and kanamycin (cat. #

K-120, Goldbio) were added at the final concentrations of 250, 40, 20, or 50 µg/ml, respectively.

## Tandem affinity purification (TAP) and LC-MS/MS analysis

The TAP tag is composed of calmodulin-binding protein, cleavage site for tobacco etch virus (TEV) protease, and an IgG-binding site of protein A. ΔtrxA Salmonella strains harboring low-copy pWSK29::TAP or pWSK29::trxA::TAP plasmids were grown to stationary phase in LB broth at 37 °C. Bacterial cells harvested by centrifugation were incubated for 1 h at 30 °C in PBS containing 1% formaldehyde (cat. # 252549, Sigma, St. Louis, MO)[46]. Cells were disrupted by sonication in lysis buffer [50 mM Tris-HCl, pH 7.4, 50 mM NaCl, 10 mM MgCl$_2$, 1 mM EDTA, 10% glycerol, 0.01% lgepal CA-630, 0.5 mM β-mercaptoethanol and 0.5 mM phenylmethylsulfonyl fluoride (PMSF)]. Cell-free extracts were incubated for 2 h at 4 °C with 500 µl of an IgG sepharose resin (cat. # 17096901, GE Healthcare, Pittsburgh, PA) while gently mixing. The resins were washed three times with 5 ml of washing buffer (10 mM Tris-HCl, pH 7.4, 50 mM NaCl and 0.01% lgepal CA-630) before adding 2 ml of TEV cleavage buffer [10 mM Tris-HCl, pH 8.0, 50 mM NaCl, 0.5 mM EDTA, 1 mM DTT and 0.01% lgepal] containing 50 U of Halo TEV Protease (cat. # G660A, Promega, Madison, WI). After 5 h of incubation at room temperature, samples were eluted with TEV cleavage buffer, and the eluants were incubated for 2 h at 4 °C with agitation with calmodulin affinity resin (cat. # 214303-52, Agilent Technologies, Cedar Creek, TX) in calmodulin binding buffer [10 mM Tris-HCl, pH 8.0, 50 mM NaCl, 1 mM magnesium acetate, 1 mM imidazole, 2 mM CaCl$_2$, 0.01% lgepal CA-630 and 10 mM β-mercaptoethanol]. Proteins were eluted with calmodulin elution buffer [10 mM Tris-HCl, pH 8.0, 50 mM NaCl, 1 mM magnesium acetate, 1 mM imidazole, 2 mM EGCA, 0.01% lgepal CA-630 and 10 mM β-mercaptoethanol] and precipitated with 20% trichloracetic acid (TCA) (cat. # A11156.30, Thermo Fisher Scientific, Grand Island, NY). Precipitates were loaded onto 4–15% NuPAGE gels (cat. # NP0321, Thermo Fisher Scientific, Grand Island, NY) in 2-[N-morpholino]-ethane sulfonic acid (MES) SDS running buffer (cat. # J62138.K2, Thermo Fisher Scientific, Grand Island, NY) and visualized with Imperial Protein Stain (cat. # 24615, Thermo Fisher Scientific, Grand Island, NY). The samples were analyzed by liquid chromatography-tandem mass spectrometry (LC-MS/MS) using an LTQ-Orbitrap mass spectrometer (model # Orbitrap Velos Pro, Thermo Fisher Scientific, Grand Island, NY) at the Mass Spectrometry and Proteomics Core at the University of Colorado School of Medicine. The mass spectrometry proteomics data have been deposited to the ProteomeXchange Consortium via the PRIDE partner repository with the dataset identifier PXD037199 and 10.6019/PXD037199. TrxA-binding partners were identified using the Mascot and Protein Prospector Program (Matrix Science, Boston, MA).

## Bacterial two-hybrid system and β-galactosidase activity

A bacterial two-hybrid system that reconstitutes interactions of the T18 and T25 subunits of adenylate cyclase (cat. # EUK001, Euromedex, Souffelweyersheim, France) was used to validate interactions of TrxA with partner proteins. The pKNT25 or pKT25 plasmids encoding fruR, galS, gntR, ompR variants, phoP, pdhR, ssrB, rpoA, or yjfQ genes were transformed together with the pUT18 or pUT18C plasmids bearing wild-type or mutant trxA alleles into E. coli BTH101. The β-galactosidase activity was measured using o-nitrophenol-β-galactoside (ONPG) (cat. # N1127, Sigma, St. Louis, MO) as previously described[47]. Enzyme activity expressed as Miller units were calculated as $A_{420} \times 1000 / [\text{time (min)} \times A_{600} \times \text{cell volume (ml)}]$.

## Overexpression and purification of proteins

For pull-down assays, TrxA partner proteins were cloned into the GST fusion pGEX-6p-1 plasmid (cat. # 28954648, GE Healthcare Life Sciences, Marlborough, MA). A cytoplasmic fragment of EnvZ encompassing residues 223-450 was cloned into the GST fusion plasmid. Full-length wild-type trxA and its point mutations were cloned as C-terminal 6His fusions into the NdeI and XhoI sites of the pET-22b(+) plasmid (cat. # 70765, Novagen, Madison, WI). All constructs and mutations were confirmed by sequence analysis. Plasmids were expressed in E. coli BL21 (DE3) (cat. # EC0114, Thermo Fisher Scientific, Grand Island, NY) or Origami B (DE3) pLysS (cat. # 70839, Novagen, Madison, WI). Briefly, cells grown in LB broth at 37 °C to an OD$_{600}$ of 0.5–0.8 were treated with 0.1 mM isopropyl-β-D-thiogalactopyranoside (IPTG). After overnight culture at 25 °C, the cells were harvested and disrupted by sonication. Cell-free extracts were obtained after 30 min centrifugation at 10,000 g. The solubility of recombinant His-tagged OmpR variants substituted with A136D, I138D, or Y230D was a bit reduced in the E. coli expression system when compared to the OmpR wildtype protein. However, we obtained enough recombinant OmpR substituted proteins from soluble fractions of E. coli hosts for usage in the in vitro CD analysis, phosphotransfer reactions, pull-down, in vitro transcription, and EMSA biochemical assays. GST and 6His-tagged fusion proteins were purified according to manufacturer's protocols using Glutathione-Sepharose 4B (cat. # 20181050, bioWORLD, Dublin, OH) and TALON metal-affinity chromatography (cat. # NC9306569, Clontech, Mountain View, CA), respectively. Further purification was done by size-exclusion chromatography on Superdex 75 (cat. # 17104401, GE Healthcare Life Sciences, Marlborough, MA). Recombinant GST-SsrBc proteins containing the C-terminal effector domain (137-212 residues)[18] were used to assay TrxA oxidoreductase activity. The GST tag was removed following culture with PreScission protease in PBS containing 10 mM DTT. The SsrBc fragment was purified by affinity chromatography as described above. After overnight incubation at 4 °C, the protein was eluted and further purified by size-exclusion chromatography on Superdex 75 (cat. # 17104401, GE Healthcare Life Sciences, Marlborough, MA). SsrBc and TrxA variants were aliquoted in a BACTRON anaerobic chamber (model Bactron I, serial # 12003309, Shel Lab, Cornelius, OR) to prevent oxidation. The purity and mass of the recombinant proteins were assessed in SDS-PAGE gels.

## Pull-down assays

Interactions between recombinant proteins were analyzed using pull-down assays[48,49]. Briefly, 2 µmoles of GST-tagged fusion proteins (i.e., bait) were incubated for 2 h with 200 µL Glutathione-Sepharose 4B beads (cat. # 20181050, bioWORLD, Dublin, OH) in 50 mM Tris-HCl buffer, pH 7.5, at 4 °C. The columns were washed with 20 bed volumes of 50 mM Tris-HCl buffer, pH 7.5, and incubated for 2 h at 4 °C with rotation in the presence of 2 µmoles of 6His-tagged TrxA or OmpR variants (i.e., prey). The columns were washed with 50 mM Tris-HCl, 50 mM NaCl buffer, pH 7.5. The proteins eluted with 50 mM Tris-HCl, 500 mM NaCl buffer, pH 7.5 were precipitated with 10% TCA. The precipitants were resuspended in SDS sample buffer and loaded onto 15% SDS-PAGE gels. 6His-tag fusion proteins were detected by immunoblot using a 1:1000 dilution of an anti-6His antibody (cat. # 600-401-382, Rockland Immunochemicals, Limerick, PA), followed by a 1:10,000 dilution of goat anti-rabbit IgG antibodies conjugated to horseradish peroxidase (cat. # 65-6120, Thermo Fisher Scientific, Grand Island, NY). The blots were processed using the ECL prime Western blotting detection reagent (cat. # RPN2232, GE Healthcare Life Sciences, Marlborough, MA) and visualized by ChemiDoc XRS imaging system (model # 1708265, Bio-Rad, Hercules, CA). Purified GST proteins served as negative controls.

## Immunoprecipitation

To assess whether intact TrxA binds to OmpR protein in vivo at physiological protein concentrations expressed from their native loci in the Salmonella chromosome, immunoprecipitation assays were performed using Dynabeads Protein G resin (cat. # 10003D, Invitrogen, Waltham, MA) according to manufacturer's protocols. Briefly,

FLAG::*ompR*-, *trxA*::6His-expressing *Salmonella* grown overnight in LB broth were disrupted by sonication after resuspension in lysis buffer (20 mM Tris-HCl, pH 8.0, 137 mM NaCl, 10% glycerol, 1% Nonidet P-40, and 2 mM EDTA). Supernatants clarified upon centrifugation for 10 min at 13,000 $g$ were treated with 10 µg of anti-FLAG monoclonal antibody (cat. # F1804, Sigma, St. Louis, MO) and 1.5 mg (50 µL) Dynabeads Protein G resin. After overnight incubation with agitation at 4 °C, tubes were placed in a DynaMag magnet, and excess antibodies were washed with PBS containing 0.02% tween 20. FLAG-tagged proteins bound to the antibodies in the columns were eluted in 450 µl of 50 mM glycine buffer, pH 2.8. The pH was immediately neutralized to 7.5 with 5 µl of 1.5 M Tris-HCl, pH 8.8.

To visualize FLAG-OmpR proteins, 5% of the elutes were separated onto 12% SDS-PAGE gels. The blots were treated with a 1:500 dilution of anti-FLAG monoclonal antibody (cat. # F1804, Sigma, St. Louis, MO), followed by a 1:20,000 dilution of horseradish peroxidase-conjugated, goat anti-mouse IgG (cat. # 45-000-692, Fisher Scientific, Grand Island, NY). To detect TrxA-6His proteins bound to FLAG-OmpR, the remaining samples were precipitated by 10% TCA. The precipitants were resuspended in SDS sample buffer, and the specimens were resolved onto 15% SDS-PAGE gels. 6His-tag TrxA proteins were detected by immunoblot using a 1:1000 dilution of an anti-6His antibody (cat. # 600-401-382, Rockland Immunochemicals, Limerick, PA), followed by a 1:10,000 dilution of horseradish peroxidase-conjugated, goat anti-rabbit IgG antibodies (Fisher Scientific). As an internal control, 10 µl of cell-free extracts were loaded onto 4-20% gradient SDS-PAGE gels (cat. # 5671094, Bio-Rad, Hercules, CA). The blots were probed with 1: 5000 of an anti-DnaK monoclonal antibody (cat. # ab69617, Abcam, Waltham, MA), followed by the addition of 1:20,000 of a goat anti-mouse IgG conjugated with horseradish peroxidase (Fisher Scientific). The blots were processed using ECL prime Western blotting detection reagent (cat. # RPN2232, GE Healthcare Life Sciences, Marlborough, MA) and visualized by G:BOX imaging system (SYNGENE, Frederick MD).

## Protein expression for NMR

For expression and purification of isotope-labeled TrxA proteins, *E. coli* BL21(DE3) harboring pET22b::TrxA plasmid grown in LB broth at 37 °C overnight was diluted 1:30 into M9 salt medium [3.37 mM $Na_2HPO_4$, 2.2 mM $KH_2PO_4$, 0.855 mM NaCl and 0.935 mM $(NH_4)_2Cl$] supplemented with 0.4% glucose, 1 mM $MgSO_4$ and 0.3 mM $CaCl_2$. $^{15}NH_4Cl$ (cat. # 299251, Sigma, St. Louis, MO) and/or $^{13}C$-glucose (cat. # 389374, Sigma, St. Louis, MO) were substituted in M9 medium to produce $^{15}N$, $^{13}C$- or $^{15}N^{13}C$-labeled TrxA proteins. Cells grown in M9 salt medium at 37 °C to an $OD_{600}$ of 0.7 were induced with 0.1 mM IPTG overnight at 25 °C. Isotope-labeled TrxA proteins were purified using TALON metal-affinity chromatography (cat. # NC9306569, Clontech, Mountain View, CA). Proteins were concentrated by ultrafiltration using a 3-kDa Amicon Ultracel membrane (cat. # UFC900324, Millipore, St. Louis, MO) with 20 mM potassium phosphate buffer, pH 6.5, containing 50 mM NaCl. 6His-tagged fusion unlabeled OmpR and PhoP proteins were overexpressed in *E. coli* Origami B (DE3) and *E. coli* BL21(DE3), respectively, and purified as described above in "overexpression and purification of proteins". OmpR and PhoP proteins were suspended in 20 mM $KPO_4$ buffer containing 50 mM NaCl, adjusted to pH 6.5 and pH 7.0, respectively. The purity of proteins was evaluated by SDS-PAGE.

For expression and purification of $^2H$,$^{13}C$,$^{15}N$-labeled OmpR proteins, the following growth pipeline was used to grow *E. coli* Origami B (DE3) pLysS bearing pET22b::OmpR plasmid in order to adapt *E. coli* from $H_2O$-based to $D_2O$ (cat. # 151882, Sigma)-based medium[50]. Briefly, cells cultured in $H_2O_2$-based LB broth at 37 °C overnight were diluted 1:20 in $D_2O$-based LB medium. After overnight incubation at 37 °C, cells were diluted 1:20 in $D_2O$-based M9 salt medium replaced with $^{15}NH_4Cl$ and $^{13}C$-glucose. After incubation at 37 °C to an $OD_{600}$ of 0.5, cells were induced by 0.1 mM IPTG overnight at 25 °C. Isotope-labeled OmpR proteins were purified using TALON metal-affinity chromatography (cat. # NC9306569, Clontech, Mountain View, CA) and were suspended in 50 mM Tris-HCl, pH 7.0, buffer containing 50 mM NaCl. The purity of proteins was assessed by SDS-PAGE.

## NMR resonance assignments and response regulator HSQC titrations

The $^{15}N$,$^{13}C$-labeled apo sample contained 1.5 mM TrxA in 20 mM $KPO_4$, 50 mM NaCl, and 3% $D_2O$ at pH 6.5. The $^2H$,$^{15}N$,$^{13}C$-labeled apo sample contained 300µM OmpR in 50 mM Tris-HCl, 50 mM NaCl, and 3% $D_2O$ at pH 7. NMR assignment experiments for TrxA were performed at 298 K on a triple-resonance Varian 600 MHz spectrometer (Agilent Technologies, Santa Clara, CA). The sequence-specific backbone assignment was determined using $^{15}N$-HSQC, HNCACB, and CBCA(co)NH. For OmpR, $^{15}N$-HSQC, $^{15}N$-TROSY-HSQC, TROSY-HNCACB, TROSY-HNcoCACB, and TROSY-HNcoCA assignment experiments were performed at 298 K on triple-resonance Varian 600 and 900 MHz spectrometers. All data were processed with NMRPipe version 10.9 (NIST IBBR)[51] and analyzed using CCPNMR version 2.4.2 (CCP)[52]. Secondary structure was determined using the Secondary Structural Propensity (SSP) score program[53], which combines different chemical shifts into a single, residue-specific SSP score. Input shifts were those of $^1H^N$, $^{15}N$, $^{13}C\alpha$, and $^{13}C\beta$. Backbone chemical shift assignments have been deposited to the Biological Magnetic Resonance Data Bank (BMRB) with the accession numbers 51150 and 51151 for TrxA and OmpR, respectively.

In order to determine the interaction interface between TrxA and OmpR or PhoP, chemical shift perturbations (CSPs) and peak intensity changes were measured using $^{15}N$-HSQC titrations. First, 300 µM $^{15}N$-labeled TrxA was incubated with 0, 75, 150, 300, and 600 µM of OmpR. In a second experiment, 100 µM $^{15}N$-labeled TrxA was incubated with 0, 50, 100, and 200 µM of PhoP. Titration experiments were performed at 298 K on a triple-resonance Varian 900 MHz spectrometer equipped with a cryoprobe (Agilent Technologies, Santa Clara, CA) in 20 mM $KPO_4$ buffer with 50 mM NaCl and 3% $D_2O$ at either pH 6.5 or 7.4 for the OmpR and PhoP titrations, respectively. CSPs of the protein upon binding was calculated using the following equation

$$\triangle \delta_{H,N}(CSP) = \sqrt{(\triangle \delta_H)^2 + 0.15(\triangle \delta_N)^2}.$$

We attempted to fit the $K_d$ value from the CSP values using GraphPad but both interactions appear to be too tight for this fast exchange assumption.

To determine the opposing interface on OmpR, 225 and 450 µM unlabeled TrxA was titrated into 150 µM $^{15}N$-OmpR, and $^{15}N$-HSQCs were measured at 298 K on a triple-resonance Varian 900 MHz spectrometer equipped with a cryoprobe (Agilent Technologies, Santa Clara, CA) in 50 mM Tris-HCl buffer with 50 mM NaCl and 3% $D_2O$ at pH 7. We did not attempt to fit the CSPs or peak intensity changes due to the low number of titration points that were measured.

## Consensus logs

To model conservation across structurally similar proteins, PDB 4HUA[32] (https://www.rcsb.org/structure/4HUA) for TrxA was uploaded into 3D-Blast (http://3d-blast.life.nctu.edu.tw/)[54,55] and searched across the PDB database. Hits with an e-value of under $10^{-10}$ were collected, and the resulting 448 sequences were aligned by Clustal Omega (https://www.ebi.ac.uk/Tools/msa/clustalo/) (Madeira 2019)[56]. The resulting aligned FASTA was uploaded to WebLogo (https://weblogo.berkeley.edu/)[57,58] and trimmed to include only residues found to interact with either OmpR or PhoP by NMR.

## CD spectroscopy

Secondary structure of TrxA or OmpR variants was analyzed by circular dichroism (CD) spectroscopy. Briefly, 0.3-0.4 mg/ml of the indicated recombinant proteins prepared in 200 µl of 50 mM Tris-HCl buffer, pH 7.4, was performed at the Biophysics Core of the University

of Colorado School of Medicine using a Jasco 815 spectrometer (Jasco, Easton, MD) with constant nitrogen flushing at 37 °C. Spectra are the average of 3 independent scans from 2 individual experiments. Binding constants were calculated with Microsoft Excel version 16.65.

## Protein binding affinity

Protein binding affinity between TrxA variants and GST-OmpR variants was evaluated by Microscale Thermophoresis (MST, Nano Temper Technologies, Műnchen, Germany) using a Monolith NT.115 (Nanotemper Inc., Munchen, DE). Briefly, 100 nM TrxA-6His variants were labeled with 2 µl NT647 fluorescent dye using the Monolith NT His-Tag Labeling Kit RED-Tris-NTA (cat. # MO-L018, Nano Temper Technologies). GST-OmpR variants were diluted in 50 mM Tris HCl buffer, pH 7.5, containing 75 mM NaCl with an equal volume of 20 nM NT647-labeled TrxA variants, and loaded into premium capillaries (cat. # MO-K025, Nano Temper Technologies, Műnchen, Germany). Binding was measured on a Monolith NT.115 at LED/excitation and 40% MST power at the Biophysics Core of the University of Colorado School of Medicine. $K_d$ values were determined from the average of 4 independent measurements from 2-3 individual experiments using the MO Affinity Analysis program of Nano Temper Technologies.

## Oxidoreductase activity of TrxA

The oxidoreductase activity of TrxA variants was assessed by in vitro detection of the redox state of SsrBc proteins containing residues 137–212[18]. Briefly, 15 µM recombinant SsrBc proteins were treated with or without 500 µM $H_2O_2$ in the presence and absence of 50 µM TrxA variant proteins for 1 h at 37 °C. Samples were loaded onto 4–20% non-reducing gradient SDS-PAGE gels (cat. # 5671094, Bio-Rad, Hercules, CA). The 9.2-kDa monomer and the 18.4-kDa dimer bands were visualized by Imperial Coomassie Brilliant Blue staining.

## Intracellular survival

J774A.1 catalog # TIB-67™ cells (American Type Culture Collection, Manassas, VA) were cultured in RPMI medium (cat. # R0883, Sigma, St. Louis, MO) supplemented with 10% heat-inactivated fetal bovine serum (cat. # 10082147, Thermo Fisher Scientific, Grand Island, NY), 1 mM sodium pyruvate (cat. # 11360070Fisher Scientific, Grand Island, NY), 2 mM L-glutamine (cat. # A2916801,Thermo Fisher Scientific, Grand Island, NY) and 20 mM HEPES (cat. # 15630080, Thermo Fisher Scientific, Grand Island, NY). Confluent J774 were infected at an MOI of 2 with *Salmonella* grown overnight in LB broth at 37 °C. Intracellular survival was assessed after cell host lysis by the addition of PBS containing 0.1% Triton X-100. The specimens were serially diluted in PBS and the *Salmonella* burden was enumerated onto LB agar plates after overnight incubation at 37 °C. Fold replication was calculated from the number of bacteria recovered after 18 h of infection compared to time zero.

## Transcriptional analysis of intracellular *Salmonella*

J774 macrophage-like cells were grown in RPMI-medium at 37 °C to ~90% confluence on 150 mm plates (cat. # 430599, Corning, St. Louis, MO) in a 5% $CO_2$ incubator. Cells were washed once with prewarmed PBS prior to the addition of fresh RPMI-medium. Macrophages were infected at an MOI of 50 with the indicated *Salmonella* grown overnight in LB at 37 °C with shaking. After 30 min incubation, cells were washed twice with prewarmed PBS, and the specimens were cultured in fresh RPMI-medium for 30 min at 37 °C in a $CO_2$ incubator. The samples were treated for 1 h with 50 µg/ml gentamicin (cat. # G1264, Sigma, St. Louis, MO). The medium was replaced with fresh RPMI-medium containing 10 µg/ml gentamicin. After 8 h of infection, the cells were washed once with prewarmed PBS and the specimens were lysed in ice-cold PBS containing 0.1% Triton X-100. The cells were scraped and vortexed for 20 sec. Cell host debris was discarded by centrifugation at 76.5–119.5 g for 5 min at room temperature. Bacteria

present in the supernatants were collected by centrifugation at 7649.2 g for 5 min followed by a wash in ice-cold PBS. Bacterial pellets were collected, and the specimens were processed for RNA isolation, cDNA synthesis, and quantitative RT-PCR as described below. DNA-free RNA was purified using a High Pure RNA isolation kit according to the manufacturer's instructions (cat. # 11828665001, Roche, Indianapolis, IN). First-strand cDNA generation was generated using 2 µg RNA, 200 U Moloney murine leukemia virus (MMLV) reverse transcriptase (cat. # M1701, Promega, Madison, WI), and 0.5 µg N6 random primers (cat. # 48190011, Thermo Fisher Scientific, Grand Island, NY). Relative nucleic acid quantitation was done using the SYBR green quantitative real-time PCR master mix (cat. # 04707516001, Roche, Indianapolis, IN) using gene-specific primers (Table S3) on an CFX Connect Real-Time System (Biorad, Hercules, CA). Data evaluation of 3 biological replicates done in duplicate was performed using the threshold cycle ($2^{-\Delta\Delta CT}$) method using Microsoft Excel. The $\Delta\Delta CT$ is calculated as $\Delta\Delta CT = \Delta CT$ mutant gene (mutant target gene CT−mutant reference gene CT)−$\Delta CT$ complementation strain gene (complementation target gene CT − complementation reference CT). Gene expression was normalized to internal levels of the housekeeping gene *rpoD* as a reference. Transcripts up- or downregulated ≥ 2-fold were considered significant.

## Animal studies

Under animal studies, 6 to 8-week-old, male and female C57BL/6 mice were inoculated i.p. with ~1000 CFU of a *Salmonella* mixture containing equal numbers of wild-type and mutant strains. The bacterial burden was quantified in livers and spleens 3 days post-infection by plating onto LB agar containing the appropriate antibiotics. Competitive index was calculated as (strain 1 ÷ strain 2)$_{output}$ ÷ (strain 1 ÷ strain 2)$_{input}$. All mice were bred on according to protocols approved by the Institutional Animal Care and Use Committee (IACUC) at the University of Colorado School of Medicine. The macroenvironment is electronically controlled to provide 22.2 ± 1 °C, a 1:10 light/dark cycle, and 30–40% humidity with at least 12 air changes per hour.

## EnvZc autophosphorylation and OmpR phosphorylation

Autophosphorylation was performed as previously described[59] with modifications. Briefly, 10 µg GST-EnvZc protein was incubated for the indicated time at room temperature in reaction buffer (25 mM Tris-HCl, pH 7.5, 50 mM KCl, and 25 mM $MgCl_2$) containing 0.75 µCi of [γ32-P]-ATP (cat. # BLU002250UC, PERKIN – ELMER, 10 Ci/mmol 2 mCi/ml Lead, 250 µCi).). Where indicated, 30 µg of OmpR variants were incubated at room temperature in a reaction buffer containing 0.75 µCi of [γ32-P]-ATP and 30 µg GST-EnvZc. Samples mixed with SDS loading buffer were analyzed in 10% (w/v) SDS-PAGE gels. The dry gels were exposed to K-screen overnight and visualized using a phosphorimager (Amersham, Buckinghamshire, England).

## Immunoblots

To assess the facilitation of TrxA proteins in *Salmonella*, the abundance of FLAG-OmpR was detected by immunoblot assay with the indicated strains. Cells grown in LB broth overnight at 37 °C were disrupted by sonication in PBS, and half of lysates were centrifuged for 10 min at 13,000 g to collect cell-free extracts. Protein concentrations of crude extracts and cell-free extracts were determined with BCA protein assay kit (cat. # 23227, Pierce, Rockford, IL) then the specimens were loaded onto 12% SDS-PAGE gels. The blots were treated with a 1:500 dilution of anti-FLAG monoclonal antibody (cat. # F1804, Sigma, St. Louis, MO), followed by a 1:20,000 dilution of goat anti-mouse IgG conjugated with horseradish peroxidase (cat. # 45-000-692, Fisher Scientific, Grand Island, NY). As an internal control, the blots were treated 1: 5000 with an anti-DnaK monoclonal antibody (cat. # ab69617, Abcam, Waltham, MA), followed by a 1:20,000 dilution of goat anti-mouse IgG conjugated with horseradish peroxidase. The specimens were visualized using the ECL

prime Western blotting detection reagent (cat. # RPN2232, GE Health-care Life Sciences, Marlborough, MA) in the ChemiDoc XRS imaging system (Bio-Rad, Hercules, CA). Protein density was measured by the ImageJ program (NIH).

## Electrophoretic mobility shift assays

We evaluated the binding of OmpR variants to a 75 bp DNA fragment containing the F2 and F3 OmpR binding sites of P*ompF*[60]. The P*ompF* promoter was amplified by PCR (Table S4) and purified using GeneJET PCR Purification Kit (cat. # K0702, Thermo Fisher Scientific, Grand Island, NY). An 80 bp of DNA template containing the flanking region of the F3 OmpR binding site was used as a negative control. The DNA templates, confirmed by sequence, were labeled with [$\gamma$-$^{32}$P]ATP (cat. # BLU002250UC, PERKIN – ELMER, 10 Ci/mmol 2 mCi/ml Lead, 250 $\mu$Ci) using T4 polynucleotide kinase (cat. # M4101, Madison, WI). The DNA probes (approximately $10^4$ cpm) were incubated at room temperature with 100 ng of recombinant OmpR variants in 50 mM Tris-HCl buffer, pH 7.5, supplemented with 50 mM KCl, 5 mM $CaCl_2$, 5% glycerol, 0.0025% Nonidet P-40, and 0.8 $\mu$g of poly (dI-dC) (cat. # 20148E, Thermo Fisher Scientific, Grand Island, NY).. After 30 min incubation, the reaction mixtures were analyzed in 5% non-denaturing poly-acrylamide gels. The gels were dried, wrapped in plastic, and exposed against K-screens overnight. K-screens were visualized using a phos-phorimager (Amersham, Buckinghamshire, England).

## In vitro transcription

In vitro transcription reactions were quantified by qRT-PCR as pre-viously described[48]. Briefly, transcription reactions were performed in 20 mM Tris-HCl, pH 7.5, 100 mM KCl, 0.1 mM EDTA, 0.05% Nonidet P-40, 200 $\mu$M of each ATP, GTP, CTP and UTP (cat. # R0481, Thermo Fisher Scientific), 8 U RiboLock RNase inhibitor (cat. # EO0382, Thermo Fisher Scientific), 1 nM *ompf* DNA templates (Table S3), 5 nM *E. coli* holoenzyme RNA polymerase (cat. # M0551S, New England Biolabs, Ipswich, MA) and 2.5 $\mu$M of OmpR variants[61]. Reactions were incubated at 37 °C for 10 min, and then heat-inactivated at 70 °C for 10 min. After DNaseI treatment, template DNA was removed from the reactions with DNA-free DNA Removal Kit (cat. # AM1906, Thermo Fisher Scientific, Grand Island, NY), and the resulting RNA was used as template to generate cDNA using 50 U MMLV reverse transcriptase (cat. # M1701, Promega, Madison, WI), 10 $\mu$M *ompf* reverse primer (Table S4) and 20 U RNase inhibitor (cat. # N2611, Promega). The amount of cDNA synthesized following 1 h of incubation at 42 °C was quantified by qRT-PCR using *ompf*-specific primers and probe (Table S4). Specific transcripts were normalized to standard curves using known amounts of transcript concentrations.

## Statistical analysis

With the exception of the NMR, all other statistical analyzes were performed using GraphPad Prism (GraphPad Software version 9.3). One-way and two-way ANOVA were performed, considering $p < 0.05$ as statistical significance. Error bars indicate mean ± SD.

## Reporting summary

Further information on research design is available in the Nature Research Reporting Summary linked to this article.

## Data availability

All data generated or analyzed during this study are included in this published article (and its supplementary information files). The NMR chemical shifts have been deposited in the Biological Magnetic Reso-nance Bank under accession codes 51150 and 51151. The mass spectrometry proteomics data have been deposited to the Proteo-meXchange Consortium via the PRIDE partner repository with the dataset identifier PXD037199 and 10.6019/PXD037199. Source data are provided with this paper.

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

## Acknowledgements

We thank Dr. Jessica Jones-Carson for kindly providing the mice and for helpful discussions, and Dr. Monika Dzieciatkowska for performing the MS analysis. This work was supported by NIH grants R01AI55493 (AVT), R01AI136520 (AVT), R01GM130694 (BV), S10 OD025020-01 (BV), P30 CA046934 (BV), and T32AI052066 (JT) and VA grants BX0002073 (AVT),

and IK6BX005384 (AVT). Dr. Vögeli's laboratory is also funded by NIH Grant, a start-up package by the University of Colorado.

## Author contributions

Conceptualization, J.S.K., A.B., J.T., B.V., and A.V.T.; Methodology, J.S.K., A.B., J.T., and A.V.T.; Bioinformatics, A.B. and J.T.; Investigations, J.S.K., A.B., L.L., M.A.H., and S.K.; Manuscript writing, J.S.K., A.B., and A.V.T.; Supervision, B.V. and A.V.T.

## Competing interests

The authors declare no competing interests.
