## [Peer review file · Nature Communications]

REVIEWER COMMENTS

Reviewer #1 (Remarks to the Author):

In this manuscript, Kim et al. demonstrated that the response regulator OmpR binds to TrxA and they argue that this interaction is critical for the transcription of the target genes, in addition to another response regulator SsrB that they reported previously (PMID: 26997275). Although the current study is not merely incremental, their arguments are not well supported by the data at this point. My concerns and comments are listed below.

General comments:

-Figure legends are too brief. So, it is hard to understand what each label indicates and how the experiments were done.

-In many places, proper citations are missing.

Specific comments:

1. The interactions of TrxA and response regulators were examined in vitro using purified proteins. It would be necessary to verify it in vivo with intact proteins at normal cellular concentrations.

2. Is there any reason for choosing D, a charged amino acid, for the amino acid substitution study? All the substitutions with D killed the protein interaction with regulators, whereas the substitutions in Cs with A did not. Could it be due to the amino acid properties? Have the authors tested C33/36D substitutions instead of that with A? Alternatively, R74A or G75A instead of R74D or G75D?

3. The defective survival/growth phenotypes of the TrxA variants (Fig. 4A) showed different patterns with the defective binding phenotypes, suggesting those two phenotypes may not be directly correlated.

This was also found for the OmpR variants: their defectiveness of survival/growth and their TrxA binding phenotypes are different, suggesting those two phenotypes are not directly correlated.

4. In vitro studies were mostly done using OmpR and PhoP proteins, yet gene expression studies were focused on SsrB-regulon, SPI-2 genes. Does PhoP- or OmpR-regulons' expression also behave in a similar way?

This also applies to the study with the OmpR variants. It could be very indirect as there are multiple factors involved in controlling SPI-2 gene expression such as SsrB, H-NS, and so on. How about the ompF gene expression instead?

5. Data in Fig. 6D are unclear. Maybe too much probe DNA was used in this assay. Was there a free probe? What is the band appearing in the D55A?

6. How do the authors interpret the soluble/insoluble OmpR variant or wild-type proteins? It is unclear how the fractionations were made due to the lack of methods, which makes it hard to interpret the data as well. Soluble fractions can vary depending on the methods such as buffer condition, lysis method, any treatment like fixation, and so on.

Also, do the TrxA variants defective in regulator interaction also cause changes in regulators' solubility?

7. It is unclear what data suggest the description in lines 274-275. The reduction of the examined target genes' expression could be due to less activation of the regulator (i.e., less phosphorylation) in addition to what the authors described.

8. Although it is clear that the TrxA variants are defective in some of TCS target genes, it remains unclear how the TrxA promotes such gene expression and how those mutations would lead to reduced gene expression.

Minor comments:

-Given that the Nat Comm is a broad-spectrum journal, briefly explaining how to read the NMR data such as CSPs would help readers' understanding.

-It would be better to show actual numbers at 0 and 18 h for the survival assay. Also, it is not clear how the 100% is defined for the % survival.

-“Enable” is a very strong argument than what the data show. Transcriptional defects were only 2-4-fold.

Reviewer #2 (Remarks to the Author):

RE: NCOMMS-22-01078-T

In this manuscript, Vázquez-Torres and co-workers report promiscuous interactions between bacterial response regulators and thioredoxin, the latter of which is encoded by the *trxA* gene in *Salmonella*. Through detailed biochemical studies, the authors demonstrate that conserved hydrophobic and charged residues on the surface of TrxA serve as a docking station for multiple members of the response regulators. Furthermore, mutation of TrxA interfacial residues abates *Salmonella* growth in macrophage-like J774 cells and reduces bacterial pathogenicity in mice. Conversely, TrxA-interacting residues of OmpR have been mapped, and their effects on TrxA-binding have been demonstrated by mutagenesis studies. Overall, this is a solid study that has provided convincing evidence to support the redox-independent function of TrxA as a chaperone for response regulators in the two-component signaling systems. However, the concept of the redox-independent chaperone function of thioredoxin has been reported nearly two decades ago (e.g., Chaperone properties of *Escherichia coli* thioredoxin and thioredoxin reductase. *Biochem J.* 2003; 371: 965–972). Furthermore, this work is mainly an extension of the published work by the same authors (*Cell Rep.* 2016 Mar 29; 14: 2901-11), in which the authors conclude that “Thioredoxin-1 binds to the flexible linker, which connects the receiver and effector domains of SsrB, thereby keeping this response regulator in the soluble fraction.” Therefore, the discoveries reported in this manuscript appear incremental and are thus unfit for publication in *Nature Communications*.

Reviewer #3 (Remarks to the Author):

The paper “Promiscuity of bacterial response regulators for thioredoxin arouses virulence programs” by Kim et al identifies thioredoxin as a potential chaperone for multiple response regulators including SsrB, OmpR, and PhoP. The authors utilize NMR to 1) identify residues in thioredoxin that mediate binding to OmpR and PhoP, and 2) identify residues in OmpR that mediate binding to thioredoxin. Based on these NMR results, the authors then construct specific point mutations to abrogate thioredoxin-response regulator binding, and show that these point mutants exhibit defects in intracellular replication in J774 cells as well as defects in mouse colonization. Overall, the paper is very well-written and the experiments are elegantly done and well controlled. I particularly like the careful validation of point mutants by the authors- for example, by making sure that the mutant proteins are folded correctly by CD *in vitro* and by making sure that the mutants are expressed at comparable levels to the wild-type protein by western blots *in vivo*. However, there are a few points that I think the authors should consider addressing prior to publication, as listed below.

Major questions:

1) Given that *Salmonella* encodes ~30 response regulators, and that response regulators generally have the same 2-domain structure, I am curious about why the pull-down experiment in Fig. 1A reveals that only certain response regulators are pulled down by thioredoxin. Is there something specific about the

sequence and/or architecture of response regulators that were pulled down that differentiates these regulators from the others? In addition, I am curious about why SsrB was not identified in this pull-down assay, if it is also known to bind to thioredoxin. Can the authors speculate?

2) In Fig. 4A, the authors show that the interface mutants of TrxA have an intracellular growth defect in J774 cells. Do these mutants also have a growth defect in a media control (ie, LB?) Or is this growth defect specific to intracellular survival/replication?

3) In Fig. 4, the authors show that thioredoxin mutants lead to decreased expression of SPI-2 genes, likely because the thioredoxin-binding response regulators OmpR, PhoP, and SsrB all regulate SPI-2 gene expression. However, as these response regulators (particularly OmpR and PhoP) regulate the expression of many more genes than just SPI-2, I think it would be interesting to see if other OmpR/PhoP regulated genes also exhibit decreased expression in these thioredoxin mutants, beyond just a few selected SPI-2 genes.

4) I'm confused by some of the results in Fig. 6. In Fig. 6F, the authors show that their tested point mutants in OmpR are insoluble. However, in Fig. 6C-E, the authors show that these same point mutants are still active in binding to RpoA and activating transcription of ompF. How were the authors able to work with these proteins in vitro if they were all insoluble? Are they soluble in E. coli during protein expression/purification but not in Salmonella?

5) The authors find that these OmpR point mutants grow worse intracellularly in J774 cells compared to wild-type (Fig. 7A). However, as these mutant OmpR proteins are insoluble (Fig 6F), is it possible that this is just due to the fact that Salmonella is being impacted by the build-up of insoluble proteins, which could also be detrimental to growth. Do these point mutants have a growth defect when grown in a media control (ie, LB?)

6) One of the major conclusions of the paper is that thioredoxin serves as a chaperone for different response regulators and may help maintain their solubility. I think that the authors should more directly test this claim by utilizing a thioredoxin deletion strain and then determining if thioredoxin-binding response regulators (ie, OmpR, SsrB, and PhoP) accumulate in the insoluble fraction when thioredoxin is missing (ie, like what they did in Fig. 6E with OmpR point mutants).

7) Another claim in the paper is that "mutations of the interfacial residues in OmpR do not affect interactions of this response regulator with the sensor kinase EnvZ." While I understand that these thioredoxin-binding residues are not at the phosphorylation site of response regulators, this statement is not tested directly. To test directly, the authors could perform in vitro phosphorylation assays with WT EnvZ and their point mutants of OmpR to show that there is no change in the rate/amount of phosphotransfer between EnvZ and OmpR.

8) Related to the above point, the authors can also add purified thioredoxin into these in vitro phosphorylation assays to test whether the addition of this chaperone changes the rate/efficiency of EnvZ-OmpR phosphotransfer. If OmpR exhibits increased stability in the presence of the thioredoxin chaperone, it could in theory also improve EnvZ-OmpR phosphotransfer.

Other questions:

- 1) In Fig. 1B, it looks like the noise in the 2-hybrid experiment is a lot higher with the cysteine mutant compared to WT thioredoxin, especially for SsrB, OmpR, and GalS. Do the authors know why this is?
- 2) The authors measure the binding affinity of OmpR to thioredoxin to be $\sim 6 \mu\text{M}$. What is the intracellular concentration of these proteins? In other words, is this $6 \mu\text{M}$ binding affinity in a physiologically relevant range?
- 3) In Fig. 5, the authors identify residues in OmpR that likely mediate binding to thioredoxin. Are these residues also present in PhoP/SsrB? Are they different in response regulators that were not pulled down by thioredoxin in Fig. 1?
- 4) Is the statement in line 210 (response regulators are notoriously insoluble) true? There have been multiple phospho-profiling papers (see Laub et al., 2007 *Methods in Enzymology*) in which all response regulators have been successfully purified from individual bacterial species, which would suggest that they are not that insoluble in general. Can the authors provide a source for their statement?

REVIEWER COMMENTS

Reviewer #1 (Remarks to the Author):

In this manuscript, Kim et al. demonstrated that the response regulator OmpR binds to TrxA and they argue that this interaction is critical for the transcription of the target genes, in addition to another response regulator SsrB that they reported previously (PMID: 26997275). Although the current study is not merely incremental, their arguments are not well supported by the data at this point. My concerns and comments are listed below.

General comments:

-Figure legends are too brief. So, it is hard to understand what each label indicates and how the experiments were done.

Response: As requested, the legends have been expanded.

-In many places, proper citations are missing.

Response: We have added more references to the revised text.

Specific comments:

1. The interactions of TrxA and response regulators were examined *in vitro* using purified proteins. It would be necessary to verify it *in vivo* with intact proteins at normal cellular concentrations.

Response: As suggested by the reviewer, we have now examined interactions of TrxA with the response regulator OmpR in *Salmonella* expressing these proteins as His or FLAG tags, respectively, from their native loci in the chromosome. **Fig. 1D** shows that TrxA and OmpR interact *in vivo* under physiological conditions. These results are consistent with the phenotypes recorded *in vivo* in the TAP screen and two-hybrid system in *Salmonella* and *E. coli*, respectively, as well as the multiple *in vitro* experiments with recombinant proteins. The new *in vivo* data have been added to the revised manuscript.

2. Is there any reason for choosing D, a charged amino acid, for the amino acid substitution study? All the substitutions with D killed the protein interaction with regulators, whereas the substitutions in Cs with A did not. I particularly like the careful validation of point mutants by the authors- for example, by making sure that the mutant proteins are folded correctly by CD *in vitro* and by making sure that the mutants are expressed at comparable levels to the wild-type protein by western blots *in vivo*. Could it be due to the amino acid properties? Have the authors tested C33/36D substitutions instead of that with A? Alternatively, R74A or G75A instead of R74D or G75D?

Response: We chose aspartate because the area of contact is large (~ 4400 Å²). We pursued substitutions that may alter the combined contribution of multiple interfacial residues in the binding area. Our rationale for choosing aspartate has been clarified in the revised document. As the reviewer comments, all tested TrxA mutants expressing the aspartate substitution fold correctly, are expressed at normal levels in *Salmonella* and, importantly, preserve the enzymatic activity of this oxidoreductase. Despite these wildtype phenotypes, aspartic acid substitutions in residues Arg⁷⁴, Gly⁷⁵, or Ala⁹⁴ and Leu⁹⁵ block interactions of TrxA with OmpR, PhoP and SsrB, while attenuating *Salmonella* virulence.

We have tested several other TrxA variants bearing aspartate substitutions in single residues, including A16D, K19D, A68D, I73D, I80D, K83D, V87D, A88D, A89D, V92D, L95D, G98D, A106D, L108D, and A109D. *Salmonella* expressing these aspartic acid variants of *trxA* grow in J774 cells to similar levels as *Salmonella* controls expressing the wildtype *trxA* allele (**Fig. S5C**). Moreover, the secondary structure of TrxA I6D or TrxA A88D is apparently normal, as it is the ability of these variants to bind to recombinant OmpR protein (**Fig. S4A, S4B and S4E**). These findings, which have been added to the revised manuscript, suggest that aspartic acid mutations do not indiscriminately disrupt TrxA chaperone function.

As requested by the reviewer, we have now tested aspartic acid substitutions of thioredoxin catalytic Cys³³ and Cys³⁶ residues. *Salmonella* expressing *trxA* C33D or *trxA* C36D grow as well as wildtype *Salmonella* in J774 cells (**Fig. S5B**). In addition, recombinant C33D or C36D TrxA variants bind to recombinant OmpR with apparently similar strength as the wildtype TrxA protein (**Fig. S4F and S4G**). These new data have been added to the revised paper.

3. The defective survival/growth phenotypes of the TrxA variants (Fig. 4A) showed different patterns with the defective binding phenotypes, suggesting those two phenotypes may not be directly correlated. This was also found for the OmpR variants: their defectiveness of survival/growth and their TrxA binding phenotypes are different, suggesting those two phenotypes are not directly correlated.

Response: As stated by the reviewer, *Salmonella* expressing *trxA* R74D or *trxA* A93D alleles grew in J774 cells to slightly higher densities than controls expressing *trxA* A73D I76D or *trxA* A94D L95D mutations. As seen in Fig. 2, the interfacial residues in TrxA's binding surface are generally shared between the two response regulators tested (i.e., OmpR and PhoP). However, some residues in TrxA are uniquely engaged by either OmpR or PhoP, suggesting that some residues may contribute to binding of some substrates but not others. For example, The R74D mutation seems to have less of an impact than G75D, G93D or A94D L95D mutations in the binding of TrxA to OmpR (**Fig. 3D**). In contrast, OmpR G75D, G93D or A94D L95D variants have the same detrimental effect in the binding of TrxA to PhoP or SsrB (**Fig. 3D**). The predominant role that specific residues play in the binding of TrxA to some clients but not others provides a rationale for why not all mutations in TrxA binding area have the same impact in *Salmonella* virulence. This idea has been presented in the revised Discussion.

The reviewer is correct that not all of the mutations in OmpR affect to a similar degree the growth of *Salmonella* in J774 cells. For instance, *Salmonella* carrying the *ompR* R182D variant seems to be more attenuated for growth in J774 cells than control strains bearing A136D, I138D or Y230D alleles (**Fig. 7A**). In addition to participating in the interfacial surface that mediates binding to TrxA (our data herein), Arg¹⁸² is involved in dimerization (PMID: 33152421). This might contribute to the stronger intracellular growth defect of *ompR* R182D *Salmonella* compared to controls expressing the *ompR* A136D, I138D or Y230D alleles. In support of this idea, recombinant OmpR R182D protein not only binds poorly to TrxA but also supports defective *ompF* *in vitro* transcription (**Fig. 6A, 6B, 6F**). In contrast, OmpR A136D, I138D or Y230D variants bind poorly to TrxA but all of them support wildtype levels of *ompF* *in vitro* transcription (**Fig. 6A, 6B, 6F**). It is for these reasons that we chose *Salmonella* strains bearing OmpR A136D, I138D or Y230D mutations for testing in mice. Importantly, *Salmonella* bearing OmpR A136D, I138D or Y230D mutations exhibited similar growth defects in J774 cells and C57BL/6 mice (**Fig. 7A, 7C**).

4. In vitro studies were mostly done using OmpR and PhoP proteins, yet gene expression studies were focused on SsrB-regulon, SPI-2 genes. Does PhoP- or OmpR-regulons'

expression also behave in a similar way? This also applies to the study with the OmpR variants. It could be very indirect as there are multiple factors involved in controlling SPI-2 gene expression such as SsrB, H-NS, and so on. How about the *ompF* gene expression instead?

Response: As requested, we have examined expression of PhoP- or OmpR-regulated genes in *Salmonella* expressing the TrxA I73D I76D or TrxA A94D L95D alleles. As described for SPI-2 genes, expression of *ompF* and *mgtA* is lower in *trxA* I73D I76D or *trxA* A94D L95D *Salmonella* compared to wildtype controls. These data are shown in revised **Fig. 4B**.

5. Data in Fig. 6D are unclear. Maybe too much probe DNA was used in this assay. Was there a free probe? What is the band appearing in the D55A?

Response: The band at the bottom of the blot is free probe (**Fig. 6E, S7H**). We have repeated the assay with lower amounts of probe, and have included a specimen with just the free probe control. The results, which are shown in **Fig. S7H**, confirm our original findings (**Fig. 6E**). The new panel in **Fig. S7H** shows that, compared to the wildtype OmpR protein, the variant expressing the D55A substitution binds differently to cognate DNA, an observation that is consistent with published data (PMID: 23175504, PMID: 24440835) This information has been clarified in the revised manuscript.

6. How do the authors interpret the soluble/insoluble OmpR variant or wild-type proteins? It is unclear how the fractionations were made due to the lack of methods, which makes it hard to interpret the data as well. Soluble fractions can vary depending on the methods such as buffer condition, lysis method, any treatment like fixation, and so on. Also, do the TrxA variants defective in regulator interaction also cause changes in regulators' solubility?

Response: As requested, the Materials and Methods sections contains more information regarding the buffer and centrifugation conditions used to fractionate the cytoplasmic extracts. We have analyzed solubility of OmpR in $\Delta trxA$ *Salmonella* (**Fig. S7I**). We have also observed lack of solubility of OmpR interfacial mutants in *Salmonella*'s cytoplasmic extracts (**Fig. 6G**). Together, these findings indicate that interactions of OmpR with thioredoxin keep maintain OmpR in the soluble fraction of the cytoplasm of *Salmonella*.

7. It is unclear what data suggest the description in lines 274-275. The reduction of the examined target genes' expression could be due to less activation of the regulator (i.e., less phosphorylation) in addition to what the authors described.

Response: The new data included in **Fig. 6C** demonstrate that OmpR A136D, OmpR I138D, OmpR R182D, or OmpR Y230D are as phosphorylatable as the wildtype OmpR protein by a C-terminal cytoplasmic fragment of the sensor kinase EnvZ. These findings suggest that the reduction in transcription recorded in *Salmonella* expressing *ompR* A136D, *ompR* I138D, *ompR* R182D, or *ompR* Y230D alleles cannot be attributed to intrinsic phosphorylation defects of these OmpR variants. This information has been added to the revised manuscript.

8. Although it is clear that the TrxA variants are defective in some of TCS target genes, it remains unclear how the TrxA promotes such gene expression and how those mutations would lead to reduced gene expression.

Response: A lower functional concentration of the response regulators in the soluble fraction in *Salmonella*'s cytoplasm could contribute to the lower transcription of target genes in *Salmonella* strains defective in *trxA* or *ompR*. This interpretation has been presented more clearly

in the revised manuscript. We have also indicated that other unexplored possibilities may also be possible.

Minor comments:

-Given that the Nat Comm is a broad-spectrum journal, briefly explaining how to read the NMR data such as CSPs would help readers' understanding.

Response: We have added more information on the Results section "*Mapping of thioredoxin interfacial residues that mediate binding to response regulators.*" In addition, we have modified the legend for Fig. 2.

-It would be better to show actual numbers at 0 and 18 h for the survival assay. Also, it is not clear how the 100% is defined for the % survival.

Response: As requested by the reviewer, the actual numbers for times 0 and 18 h after infection of J774 cells have been shown in revised Fig. 4A, 7A, S5B and S5C.

-"Enable" is a very strong argument than what the data show. Transcriptional defects were only 2-4-fold.

Response: As suggested, we have changed the word "enable" for "facilitate."

Reviewer #2 (Remarks to the Author):

RE: NCOMMS-22-01078-T

In this manuscript, Vázquez-Torres and co-workers report promiscuous interactions between bacterial response regulators and thioredoxin, the latter of which is encoded by the *trxA* gene in *Salmonella*. Through detailed biochemical studies, the authors demonstrate that conserved hydrophobic and charged residues on the surface of TrxA serve as a docking station for multiple members of the response regulators. Furthermore, mutation of TrxA interfacial residues abates *Salmonella* growth in macrophage-like J774 cells and reduces bacterial pathogenicity in mice. Conversely, TrxA-interacting residues of OmpR have been mapped, and their effects on TrxA-binding have been demonstrated by mutagenesis studies. Overall, this is a solid study that has provided convincing evidence to support the redox-independent function of TrxA as a chaperone for response regulators in the two-component signaling systems.

Response: We appreciate the positive comments.

However, the concept of the redox-independent chaperone function of thioredoxin has been reported nearly two decades ago (e.g., Chaperone properties of *Escherichia coli* thioredoxin and thioredoxin reductase. *Biochem J.* 2003; 371: 965–972). Furthermore, this work is mainly an extension of the published work by the same authors (*Cell Rep.* 2016 Mar 29; 14: 2901-11), in which the authors conclude that "Thioredoxin-1 binds to the flexible linker, which connects the receiver and effector domains of SsrB, thereby keeping this response regulator in the soluble fraction." Therefore, the discoveries reported in this manuscript appear incremental and are thus unfit for publication in *Nature Communications*.

Response: We did not mean to claim that this is the first instance a chaperone function has been ascribed to thioredoxin. This important paper as well as others (references 43 and 44 in main text) were cited in the original submission. For clarity, we now cite the *Biochem J* 2003 paper in other sections of the revised manuscript as well.

The novelty of our investigations lays in the discovery that several response regulators share binding with a common target (i.e., thioredoxin). This sharply contrasts with the narrow interactions that response regulators engage with their cognate sensor kinase, thereby limiting crosstalk among these central signaling prokaryotic pathways. We believe our molecular characterization of the promiscuous docking of different classes of response regulators on a large hydrophobic patch on thioredoxin surface area is a rather important discovery that should be of the interest to the broad audience targeted by *Nature Communications*.

In addition, our current characterization of the thioredoxin's chaperone function is not merely incremental of published research. Our biophysical, genetic, biochemical and functional approaches made possible direct testing of thioredoxin's chaperone interfacial residues, in isolation of its canonical oxidoreductase activity, in *Salmonella* pathogenesis. This novel aspect of our research is more clearly presented in paragraph starting at line 175. Similarly, our multifaceted approaches identified OmpR interfacial residues that do not affect interactions of this critical response regulator with cognate sensor kinase, DNA or RNA polymerase. This research enabled testing of the unique contribution that binding of OmpR to TrxA plays in *Salmonella* virulence. In our opinion, this knowledge should be of interest to a general audience.

Reviewer #3 (Remarks to the Author):

The paper "Promiscuity of bacterial response regulators for thioredoxin arouses virulence programs" by Kim et al identifies thioredoxin as a potential chaperone for multiple response regulators including SsrB, OmpR, and PhoP. The authors utilize NMR to 1) identify residues in thioredoxin that mediate binding to OmpR and PhoP, and 2) identify residues in OmpR that mediate binding to thioredoxin. Based on these NMR results, the authors then construct specific point mutations to abrogate thioredoxin-response regulator binding, and show that these point mutants exhibit defects in intracellular replication in J774 cells as well as defects in mouse colonization. Overall, the paper is very well-written and the experiments are elegantly done and well controlled. I particularly like the careful validation of point mutants by the authors- for example, by making sure that the mutant proteins are folded correctly by CD in vitro and by making sure that the mutants are expressed at comparable levels to the wild-type protein by western blots in vivo.

Response: We thank the reviewer for the kind comments.

However, there are a few points that I think the authors should consider addressing prior to publication, as listed below.

Major questions:

1) Given that *Salmonella* encodes ~30 response regulators, and that response regulators generally have the same 2-domain structure, I am curious about why the pull-down experiment in Fig. 1A reveals that only certain response regulators are pulled down by thioredoxin. Is there something specific about the sequence and/or architecture of response regulators that were pulled down that differentiates these regulators from the others? In addition, I am curious about why SsrB was not identified in this pull-down assay, if it is also known to bind to thioredoxin. Can the authors speculate?

Response: TrxA binds to several classes of response regulators within the helix-turn-helix and helix-hairpin subfamilies, suggesting promiscuity in thioredoxin's clients. Nonetheless, as the

reviewer comments, the pull-downs identified a few of the 30 response regulators encoded in the *Salmonella* genome (**Fig. 1A**). The lack of representation of some of the response regulators may reflect molecular architectures not recognized by thioredoxin, as inferred by the reviewer's comment. However, some of the response regulators may not have been identified in our screen because of underrepresentation under the experimental conditions tested. A case in point, NarL and SsrB are not likely to be expressed in *Salmonella* grown aerobically in LB broth (conditions that were used in our screen), because NarL is a regulator of nitrate anaerobic respiration and SsrB is induced under nutritional starving conditions and low pH. Nonetheless, recombinant SsrB and NarL proteins are pull-down by TrxA (**Fig. 1C** and **S1E**). The limitations of our screen conditions have been identified in the revised manuscript.

2) In Fig. 4A, the authors show that the interface mutants of TrxA have an intracellular growth defect in J774 cells. Do these mutants also have a growth defect in a media control (ie, LB?) Or is this growth defect specific to intracellular survival/replication?

Response: As requested by the reviewer, we have tested growth of *Salmonella* expressing several *trxA* interfacial mutants in LB broth (**Fig. S5A**). The growth of these *trxA* *Salmonella* mutants in LB broth is comparable to wildtype *Salmonella*, indicating that the growth defects of *Salmonella* expressing defective thioredoxin interfacial variants is relevant to intracellular conditions encountered by this enteropathogen in host cells (**Fig. 4A, 4C**). This important information has been added to the revised manuscript.

3) In Fig. 4, the authors show that thioredoxin mutants lead to decreased expression of SPI-2 genes, likely because the thioredoxin-binding response regulators OmpR, PhoP, and SsrB all regulate SPI-2 gene expression. However, as these response regulators (particularly OmpR and PhoP) regulate the expression of many more genes than just SPI-2, I think it would be interesting to see if other OmpR/PhoP regulated genes also exhibit decreased expression in these thioredoxin mutants, beyond just a few selected SPI-2 genes.

Response: As requested, we have examined *ompF* and *mgtA* gene transcription in intracellular *Salmonella* expressing different *trxA* alleles. The results show that transcription of these OmpR- or PhoP-regulated genes is also defective in *Salmonella* bearing mutations in key TrxA interfacial residues (**Fig. 4B**).

4) I'm confused by some of the results in Fig. 6. In Fig. 6F, the authors show that their tested point mutants in OmpR are insoluble. However, in Fig. 6C-E, the authors show that these same point mutants are still active in binding to RpoA and activating transcription of *ompF*. How were the authors able to work with these proteins *in vitro* if they were all insoluble? Are they soluble in *E. coli* during protein expression/purification but not in *Salmonella*?

Response: The solubility of recombinant His-tagged OmpR variants substituted with A136D, I138D or Y230D was a bit reduced in the *E. coli* system. However, we obtained enough recombinant OmpR proteins from soluble fractions of *E. coli* hosts for usage in the *in vitro* CD analysis, phosphotransfer reactions, pull-down, *in vitro* transcription, and EMSA biochemical assays. This information has been clarified in the "Overexpression and purification of proteins" section of the Materials and Methods. The loss of solubility of OmpR variants bearing A136D, I138D or Y230D substitutions was mostly noted when the proteins were expressed naturally by *Salmonella* (**Fig. 6G**). Similarly, wildtype OmpR protein fractionated with the insoluble fraction in Δ *trxA* *Salmonella*. We propose that thioredoxin is necessary to keep OmpR soluble under the

crowded conditions of the cytoplasm of *Salmonella*. This information has been clarified in the revised manuscript.

5) The authors find that these OmpR point mutants grow worse intracellularly in J774 cells compared to wild-type (Fig. 7A). However, as these mutant OmpR proteins are insoluble (Fig 6F), is it possible that this is just due to the fact that *Salmonella* is being impacted by the build-up of insoluble proteins, which could also be detrimental to growth. Do these point mutants have a growth defect when grown in a media control (ie, LB?)

Response: This is an interesting idea. However, the new analysis included in the revised manuscript indicates that *Salmonella* expressing *ompR A136D*, *ompR I138D* or *ompR Y230D* alleles grow in LB broth as well as wildtype controls (Fig. S7K).

6) One of the major conclusions of the paper is that thioredoxin serves as a chaperone for different response regulators and may help maintain their solubility. I think that the authors should more directly test this claim by utilizing a thioredoxin deletion strain and then determining if thioredoxin-binding response regulators (ie, OmpR, SsrB, and PhoP) accumulate in the insoluble fraction when thioredoxin is missing (ie, like what they did in Fig. 6F with OmpR point mutants).

Response: As suspected, SsrB protein partitions in the insoluble fraction in Δ *trxA* *Salmonella* (PMID: 26997275). This information has been clarified in the revised manuscript. In addition, OmpR segregates in the insoluble fraction in Δ *trxA* *Salmonella* (Fig. S7I).

7) Another claim in the paper is that “mutations of the interfacial residues in OmpR do not affect interactions of this response regulator with the sensor kinase EnvZ.” While I understand that these thioredoxin-binding residues are not at the phosphorylation site of response regulators, this statement is not tested directly. To test directly, the authors could perform in vitro phosphorylation assays with WT EnvZ and their point mutants of OmpR to show that there is no change in the rate/amount of phosphotransfer between EnvZ and OmpR.

Response: As requested, we have tested interactions of OmpR variants with a cytoplasmic fragment of EnvZ, as evidenced by phosphotransfer reactions. The new data demonstrate that all OmpR variants tested are phosphorylated by a recombinant EnvZ cytoplasmic fragment as efficiently as an OmpR wildtype control. This new information is shown in Fig. 6C, S7D, S7E and S7F.

8) Related to the above point, the authors can also add purified thioredoxin into these in vitro phosphorylation assays to test whether the addition of this chaperone changes the rate/efficiency of EnvZ-OmpR phosphotransfer. If OmpR exhibits increased stability in the presence of the thioredoxin chaperone, it could in theory also improve EnvZ-OmpR phosphotransfer.

Response: as discussed above, we obtained soluble recombinant OmpR A136D, OmpR I138D, OmpR R182D and OmpR Y230D proteins for the *in vitro* conditions used in our investigations. Therefore, we did not test the effects that thioredoxin has on phosphotransfer *in vitro* reactions.

Other questions:

1) In Fig. 1B, it looks like the noise in the 2-hybrid experiment is a lot higher with the cysteine

mutant compared to WT thioredoxin, especially for SsrB, OmpR, and GalS. Do the authors know why this is?

Response: It is true that the data in the two-hybrid system are more spread out in the pUT18-*trxA* C33A C36A construct compared to pUT118-*trxA*. However, a two-way ANOVA analysis indicates that with the exception of the *galS* construct, which has a $p < 0.05$, all other comparisons between pUT118-*trxA* and pUT18-*trxA* C33A C36A are not significant. This point has been clarified Results and Figure Legend of the revised document.

2) The authors measure the binding affinity of OmpR to thioredoxin to be ~6 μM . What is the intracellular concentration of these proteins? In other words, is this 6 μM binding affinity in a physiologically relevant range?

Response: The estimated concentration of OmpR in the cytoplasm of *E. coli* is 6 μM (PMID: 11973328). Therefore, the measured binding of OmpR with thioredoxin is within the biologically relevant concentration of this response regulator. This information has been added to the revised paper.

3) In Fig. 5, the authors identify residues in OmpR that likely mediate binding to thioredoxin. Are these residues also present in PhoP/SsrB? Are they different in response regulators that were not pulled down by thioredoxin in Fig. 1?

Response: As discussed above, the lack of interactions between some of the response regulators and thioredoxin (**Fig. 1A**) may reflect the absence of particular response regulators in the cytoplasm of *Salmonella* under the experimental conditions tested. It is for this reason that we prefer not to discuss this subject at the present time. Having said that, V137, I138, and Y230 are conserved in PhoP, while L228 is conserved in SsrB. We have obtained excellent peaks in our ongoing NMR analysis of recombinant PhoP protein, and are in the process of identifying the residues contributing to the PhoP/TrxA interface. At completion of our new NMR analysis, we will be in a better position to discuss common traits in response regulators that make possible their binding to TrxA.

4) Is the statement in line 210 (response regulators are notoriously insoluble) true? There have been multiple phospho-profiling papers (see Laub et al., 2007 Methods in Enzymology) in which all response regulators have been successfully purified from individual bacterial species, which would suggest that they are not that insoluble in general. Can the authors provide a source for their statement?

Response: The reviewer is correct that several response regulators are soluble, as we have experienced with OmpR and PhoP. I guess we were biased by the poor solubility of full-length recombinant SsrB protein, a response regulator the Vazquez-Torres lab has been studying for decades. The sentence has been edited following the insightful comment by the reviewer.

REVIEWER COMMENTS

Reviewer #1 (Remarks to the Author):

The manuscript by Kim et al. has been improved. Readers would appreciate their careful evaluation of variants used in this study. Most of my concerns have been resolved satisfactorily except for one (item #1). In addition to that, just minor comments regarding item #4.

1. The interactions of TrxA and response regulators were examined in vitro using purified proteins. It would be necessary to verify it in vivo with intact proteins at normal cellular concentrations.

Response: As suggested by the reviewer, we have now examined interactions of TrxA with the response regulator OmpR in Salmonella expressing these proteins as His or FLAG tags, respectively, from their native loci in the chromosome. Fig. 1D shows that TrxA and OmpR interact in vivo under physiological conditions. These results are consistent with the phenotypes recorded in vivo in the TAP screen and two-hybrid system in Salmonella and E. coli, respectively, as well as the multiple in vitro experiments with recombinant proteins. The new in vivo data have been added to the revised manuscript.

It is great to see that the authors did co-IP analysis, yet the data shown in the manuscript is not satisfying. It is surprising that the authors did not include any controls in this analysis, which is a fundamental finding of their study. They should include at least strains containing one of those tagged-protein-encoding genes (trxA-his6 alone and ompR-FLAG alone). These controls must be included, otherwise it is hard to see what those two bands mean. (It would be nice to include His6-tagged-TrxA G93D or A94D L95D, which showed reduced interaction as well as phenotypical defects.)

4. In vitro studies were mostly done using OmpR and PhoP proteins, yet gene expression studies were focused on SsrB-regulon, SPI-2 genes. Does PhoP- or OmpR-regulons' expression also behave in a similar way? This also applies to the study with the OmpR variants.

It could be very indirect as there are multiple factors involved in controlling SPI-2 gene expression such as SsrB, H-NS, and so on. How about the ompF gene expression instead?

Response: As requested, we have examined expression of PhoP- or OmpR-regulated genes in Salmonella expressing the TrxA I73D I76D or TrxA A94D L95D alleles. As described for SPI-2 genes, expression of ompF and mgtA is lower in trxA I73D I76D or trxA A94D L95D Salmonella compared to wildtype controls. These data are shown in revised Fig. 4B.

It is interesting that the ompF gene behaves differently from those others. It would be worth to discussing them. This makes sense as SPI-2 genes are induced in so-called "SPI-2 condition" whereas ompF gene expression decreases in SPI-2 condition.

One thing, it is strange to see that the *mgtA* gene behaves similarly because the *mgtA* gene is known to be not really induced (PMID: 22699622; 26561851), unlike other PhoP-induced genes (PMID: 22699622; 26561851).

Reviewer #3 (Remarks to the Author):

The authors have done a very nice job of addressing my concerns.

Reviewer #1 (Remarks to the Author):

The manuscript by Kim et al. has been improved.

Response: We appreciate the reviewer's support.

Readers would appreciate their careful evaluation of variants used in this study. Most of my concerns have been resolved satisfactorily except for one (item #1). In addition to that, just minor comments regarding item #4. It is great to see that the authors did co-IP analysis, yet the data shown in the manuscript is not satisfying. It is surprising that the authors did not include any controls in this analysis, which is a fundamental finding of their study. They should include at least strains containing one of those tagged-protein-encoding genes (*trxA*-his6 alone and *ompR*-FLAG alone). These controls must be included, otherwise it is hard to see what those two bands mean. (It would be nice to include His6-tagged-TrxA G93D or A94D L95D, which showed reduced interaction as well as phenotypical defects.)

Response: We apologize for the omission of these important controls. As requested by the reviewer, we have performed the proposed experiments and now include control strains expressing *trxA*-6His alone and FLAG-*ompR* alone (Fig. 1D). The new data show the pull-down of TrxA with the response regulator OmpR in *Salmonella* expressing these proteins from their native loci in the chromosome. The pulldown of TrxA does not occur in the absence of OmpR. These results are consistent with the phenotypes recorded *in vivo* in the TAP screen and two-hybrid system in *Salmonella* and *E. coli*, respectively, as well as the multiple *in vitro* experiments with recombinant proteins. Defective interactions between TrxA G93D or A94D L95D variants with OmpR, PhoP and SsrB is demonstrated by the bacterial two-hybrid system in Fig. 3D. The MST analysis with recombinant TrxA A94D L95D and TrxA G93D and OmpR proteins (Fig. 3C, Fig. S4E) supports the idea that mutations in TrxA interfacial residues interfere with binding to OmpR.

It is interesting that the *ompF* gene behaves differently from those others (*sifA*, *ssaV*, *mgtA*). It would be worth to discussing them. This makes sense as SPI-2 genes are induced in so-called "SPI-2 condition" whereas *ompF* gene expression decreases in SPI-2 condition.

Response: According to the research published in PMID 26561851, the reviewer is correct that SPI2 genes appear to be expressed at higher levels than the OmpR-regulated *ompF* gene by intracellular *Salmonella*. Nonetheless, Fig. 6 of manuscript PMID 26561851 shows that intracellular *Salmonella* seem to upregulate *ompF* expression in macrophages. Our data in Fig. 4B show that *Salmonella* strains expressing a deleted *trxA* gene or *trxA* chaperone mutants express lower levels of *ompF* in J774 cells compared to wildtype *Salmonella* controls. These data are consistent with the idea that the interaction of TrxA with OmpR facilitates expression of OmpR-regulated target genes.

One thing, it is strange to see that the *mgtA* gene behaves similarly because the *mgtA* gene is known to be not really induced (PMID: 22699622; 26561851), unlike other PhoP-induced genes (PMID: 22699622; 26561851).

Response: We appreciate the reviewer's comment. Our results show that the expression of *mgtA* is lower in *Salmonella* bearing TrxA chaperone mutants compared to wildtype *Salmonella*. Our analysis cannot differentiate whether the inhibition of *mgtA* in *trxA* mutants reflects effects on basal or induced *mgtA* transcription.

Reviewer #3 (Remarks to the Author):

The authors have done a very nice job of addressing my concerns.

Response: We appreciate the reviewer's support.